# The gut contractile organoid for studying the gut motility regulated by coordinating signals between interstitial cells of Cajal and smooth muscles

Rei Yagasaki[1†], Ryo Nakamura[1‡], Yuuki Shikaya[1§], Ryosuke Tadokoro[1#], Ruolin Hao[2,3], Zhe Wang[2,3], Mototsugu Eiraku[2,3], Masafumi Inaba[1], Yoshiko Takahashi[1]*

[1]Department of Zoology, Graduate School of Science, Kyoto University Sakyo-ku, Kyoto, Japan; [2]Institute for Life and Medical Sciences, Kyoto University, Sakyo-ku, Kyoto, Japan; [3]Department of Polymer Chemistry, Graduate School of Engineering, Kyoto University, Kyoto, Japan

*For correspondence: yotayota@develop.zool.kyoto-u.ac.jp

Present address: [†]Department of Molecular, Cellular and Developmental Biology, University of Michigan, Ann Arbor, United States; [‡]Ushimado Marine Institute, Okayama University, Okayama, Japan; [§]Graduate School of Science, University of Hyogo, Hyogo, Japan; [#]Department of Bioscience, Okayama University of Science, Okayama, Japan

Competing interest: The authors declare that no competing interests exist.

## eLife Assessment

This **valuable** study reports the development of a novel organoid system for studying the emergence of autorhythmic gut peristaltic contractions through the interaction between interstitial cells of Cajal and smooth muscle cells. The authors further utilized the system to provide **convincing** evidence for a previously unappreciated potential role for smooth muscle cells in regulating the firing rate of interstitial cells of Cajal. The work will be of interest to those studying development and physiology of the gut.

**Abstract** The gut undergoes peristaltic movements regulated by intricate cellular interactions. How these interactions emerge in the developing gut remains poorly explored due to a lack of model system. We here developed a novel contractile organoid that is derived from the muscle layer of chicken embryonic hindgut. The organoid contained smooth muscle cells (SMCs) and interstitial cells of Cajal (ICCs; pacemaker) with few enteric neurons and underwent periodic contractions. The organoid was formed by self-organization with morphological arrangements of ICCs (internal) and SMCs (peripheral), allowing identification of these cells in live. GCaMP-Ca$^{2+}$ imaging analyses revealed that Ca$^{2+}$ transients between ICC-ICC, SMC-SMC, or SMC-ICC were markedly coordinated. Pharmacological studies further suggested a role of gap junctions in ICC-to-SMC signaling, and also possible mechanical feedback from SMC's contraction to ICC's pace-making activities. In addition, two organoids with different rhythms became synchronized when mediated by SMCs, unveiling a novel contribution of SMCs to ICC's pace-making. The gut contractile organoid developed in this study offers a useful model to understand how the rhythm coordination between/among ICCs and SMCs is regulated and maintained during gut development.

## Introduction

In the gut, ingested material is conveyed properly along the gut axis by the gut movements called peristalsis, which is recognized as wave-like propagation of a local constriction. The physiology of gut peristalsis has extensively been studied in adults, where the peristalsis plays pivotal roles in effective transportation and digestion/absorption of inter-luminal contents. And many cases of gut-related

pathology are associated with peristaltic dysfunctions. The gut peristaltic movements are achieved by intricate regulations of intercellular functions among multiple cell types. At the site of origin of peristaltic waves (OPW), a local constriction emerges along the circumferential axis, and this is soon followed by a progressive wave of the contraction along the gut axis. During these processes, the circumferential constriction demands multiple smooth muscle cells (SMCs) to achieve simultaneous contraction. However, due to the complex structure of the gut, how such synchronization/coordination in SMCs is regulated remains largely undetermined.

It is known in vertebrates that the embryonic gut undergoes peristaltic movements even without experience of food intake. The embryonic gut, therefore, serves as a powerful model to study the intrinsic mechanisms underlying the peristalsis, contrasting with the adult gut where ingested content influences the gut motility, increasing complexity in analyses. We have recently reported using chicken embryos that sites of OPW are randomly distributed along the gut axis at early stages, and they later become confined to specific sites, and that this confinement of OPWs enables rhythmic and patterned peristaltic movements (*Shikaya et al., 2022*). One of the long-standing and important questions is how the synchronized/coordinated contraction is achieved and maintained.

In a long history of gut motility studies, it has been known that enteric nervous system (ENS), SMCs, and interstitial cells of Cajal (ICCs) play important roles in peristaltic movements (*Barajas-López et al., 1989*; *Camborová et al., 2003*; *Chevalier et al., 2020*; *Huizinga et al., 1995*; *Kito et al., 2005*; *Liu et al., 1998*; *Rumessen and Thuneberg, 1996*; *Sanders et al., 1991*; *Takaki, 2003*; *Takayama et al., 2002*; *Thomsen et al., 1998*). It has widely been accepted that: (1) at early embryonic stages, gut movement/peristalsis does not require ENS activity, (2) ICCs serve as a pacemaker dictating their rhythm to SMCs, (3) the contraction is executed by SMC and not by ICCs, since thick fibers of myosin are found solely in SMCs. A series of elaborate electrophysiological studies showed that slow waves (a type of changes in membrane potential characteristic of gut movements) occur spontaneously in ICCs but not in SMCs, and that these slow waves lead to voltage-dependent $Ca^{2+}$ influx evoking action potential, which is somehow transmitted to SMCs to execute gut contraction in register with ICC's pace-making rhythm (*Baker et al., 2021*; *Torihashi et al., 2002*). The knowledge that ICCs act as a pacemaker was supported by compelling evidence obtained by c-Kit-deficient mouse mutants (W/Wᵛ), in which ICC differentiation was severely affected leading to a failure of gut peristalsis (*Huizinga et al., 1995*; *Torihashi et al., 1999*). However, it remains largely unexplored how the intrinsic/spontaneous rhythm in a single ICC becomes synchronized among multiple ICCs which constitute intricate networks in the gut. It has also been under debate to what extent the gap junction contributes to cell-cell communications between ICCs, SMCs, or ICC-SMC. One reason is that it has been difficult to stably maintain SMCs and ICCs in cell culture conditions, and also to distinguish ICCs from SMCs in the living gut (these two types of cells originate from the same progenitor of splanchnopleural mesoderm during development). Thus, a novel model system has been awaited to circumvent these obstacles. Recently, it was reported that differentiation states of mouse hindgut-derived cells were successfully maintained for a relatively long period of time in a serum-free culture medium (*Wang et al., 2018*). In that study, many types of gut-derived cells, including not only ENS, ICC, SMCs, but also glial cells and serosa, were observed, and this large and complex cell mass was seen to undergo rhythmic contractions in vitro.

It has widely been appreciated that organoids can serve as a powerful model and tool to circumvent such obstacles of organ complexity. Relatively simple structures of organoids allow analyses at higher resolution than in vivo and also permit analyses of cell behaviors in three-dimensional (3D) environment, which might be different from behaviors in vitro confined to two dimensions. In the current study, we have developed a novel organoid called 'gut contractile organoid' by culturing chicken hindgut-derived cells in a serum-free medium. The gut contractile organoid undergoes periodic contractions, and it is essentially composed of ICCs and SMCs, the former residing centrally whereas the latter peripherally, allowing distinction between the two cell types in living organoids. These advantages enabled GCaMP-live imaging of $Ca^{2+}$ dynamics and revealed coordinated oscillations of $Ca^{2+}$ transients between ICC-ICC, SMC-SMC, and SMC-ICC. Pharmacological studies further suggested a role of gap junctions in an ICC-to-SMC signaling, and also a possible mechanical feedback from SMC's contractions to ICC's pace-making activities. In addition, by regarding an organoid as a single oscillator unit and by placing these oscillators separately in a hydrogel mold, we found that two oscillators with different rhythms became synchronized when mediated by SMCs, supporting the

notion of SMC's contribution to ICC's pace-making. The gut contractile organoid developed in this study must be useful to unveil the intrinsic mechanisms underlying the rhythm coordination and its maintenance between/among ICCs and SMCs during peristaltic movements in the embryonic gut.

## Results

### Spheroids were formed from muscle layer-derived cells of the embryonic hindgut

To develop a culture condition that would facilitate analyses of gut motility, we dissected the muscle layer (also called *tunica muscularis*) from the hindgut of chicken embryos of embryonic day 15 (E15) by removing the serosa and intestinal epithelium (mucosa) (*Figure 1—figure supplement 1*). The isolated muscle layer was dissociated into single cells to prepare $5.0 \times 10^5$ cells per culture dish. We started analyses with a culture condition with FBS-free medium and Matrigel as substrate as previously described for cultures in mice of gastrointestinal cells including serosa (*Wang et al., 2018*). We tested three kinds of FBS-free media: DMEM/Ham's F-12, Ham's F-12, and Neurobasal media (see Materials and Methods). When cultured in DMEM/Ham's F-12 or Ham's F-12, dissociated cells formed very small aggregates containing several cells at day 1, the morphology of which did not change significantly until day 5 (*Figure 1A*). In clear contrast, when cultured in the Neurobasal medium, cells formed clusters that were interconnected by elongated cells with neighboring clusters as early as day 1. These clusters grew as larger aggregates by day 3 and became spherical by day 5. Such spheroids were not observed in the conditions with Ham's F-12 or DMEM/Ham's F-12. We also tested different substrates, Poly-Lysine or collagen for dish coating with the Neurobasal medium, but neither one yielded spheroid. To know how the spherical aggregates were formed under the condition of Neurobasal medium and Matrigel, we obtained time-lapse images at two-hour intervals. Originally, sparse clusters that were loosely connected by elongated cells merged with each other, forming progressively larger clusters (*Figure 1B*, *Figure 1—video 1*). In the following experiments, we focused on the spheroids formed under the condition of Neurobasal medium and Matrigel coating.

### The gut muscle layer-derived spheroids displayed periodic contractions

The spheroids underwent reiterative contractions at day 3, and these contractions were observed at least until day 7 of culture (*Figure 2A* for normalized intensities of three representative organoids for each stage, *Figure 2—video 1*). Time-lapse imaging of the contractions combined with quantitative assessments by MATLAB (MathWorks, see Materials and methods) showed that interval periods between two successive peaks were 13.3 s, 15.4 s and 19.6 s for cultures at day 3, day 5, and day 7, respectively (median values; *Figure 2B*; contraction intervals).

### The contracting spheroid was composed of ICCs and SMCs

To determine the cell types comprising the spheroids, we performed immunostaining with antibodies against the chicken c-Kit protein for ICCs (*Yagasaki et al., 2022*), and αSMA and desmin for SMCs. It is known that ICCs and SMCs are derived from common progenitors which are c-Kit$^+$/αSMA$^+$, and that after differentiation ICCs and SMCs are c-Kit$^+$/αSMA$^-$ and c-Kit$^-$/αSMA$^+$, respectively (*Duband et al., 1993*; *Kluppel et al., 1998*). In the clusters at day 3, c-Kit$^+$/αSMA$^-$ signals were detected in the internal region, and c-Kit$^+$/αSMA$^+$ signals at the periphery (*Figure 3A*, Day 3), suggesting that internal cells were differentiated ICCs, whereas peripheral cells were ICC/SMC progenitors. At day 5 onward, c-Kit$^+$/αSMA$^-$ cells and c-Kit$^-$/αSMA$^+$ cells were segregated, which were located internally and peripherally, respectively (*Figure 3A*, Day 5), and this spatial segregation of the cells remained unchanged until day 7 (*Figure 3A*, Day 7), the latest stage examined in this study. The peripherally lining cells were also positive for desmin staining, supporting that these cells were SMCs (*Figure 3B*). Infection with RCAS-GapEGFP into a forming spheroid further visualized multipolar ICCs in the internal region (*Figure 3C*), a morphology known to be characteristic of ICC-MY which would normally reside in the layer of myenteric plexus (*Mei et al., 2009*; *Sanders et al., 2014*). ICCs of a different type were also recognized, which were elongated, thin, and lining underneath the peripheral SMCs (*Figure 3A*, Day 7). It is possible that these were ICC-IM, known to be embedded in and tightly associated with smooth muscles in the gut (*Huizinga et al., 2011*; *Iino and Horiguchi, 2006*). Unexpectedly, the spheroid contained few neural cells (ENS), if any, revealed by anti-Tuj1 antibody (*Figure 3—figure supplement*

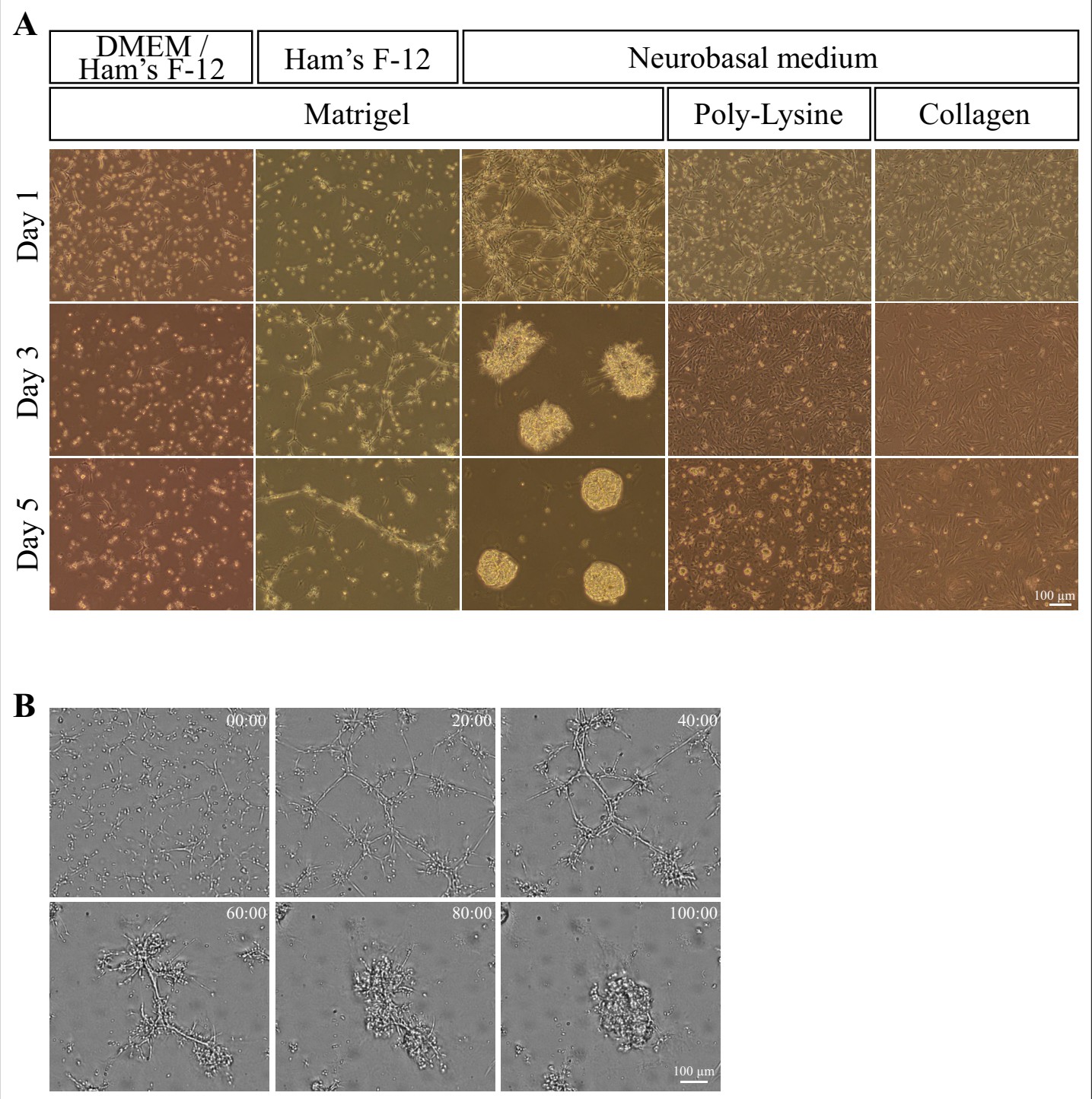

**Figure 1.** Culture of muscle layer-derived cells prepared from embryonic hindgut. (**A**) Culture of muscle layer-derived cells prepared from embryonic hindgut with FBS free-media and substrates. (**B**) Long-term time-lapse imaging after seeding on Matrigel with Neurobasal media. The images show the ability of these cells to self-assemble at 0, 20, 40, 60, 80, and 100 hours taken from *Figure 1—video 1*. Scale bars: 100 µm (**A**, **B**).

The online version of this article includes the following video and figure supplement(s) for figure 1:

**Figure supplement 1.** Dissection and layers of the E15 chicken embryonic hindgut.

**Figure 1—video 1.** Long-term time-lapse imaging after seeding.

https://elifesciences.org/articles/97860/figures#fig1video1

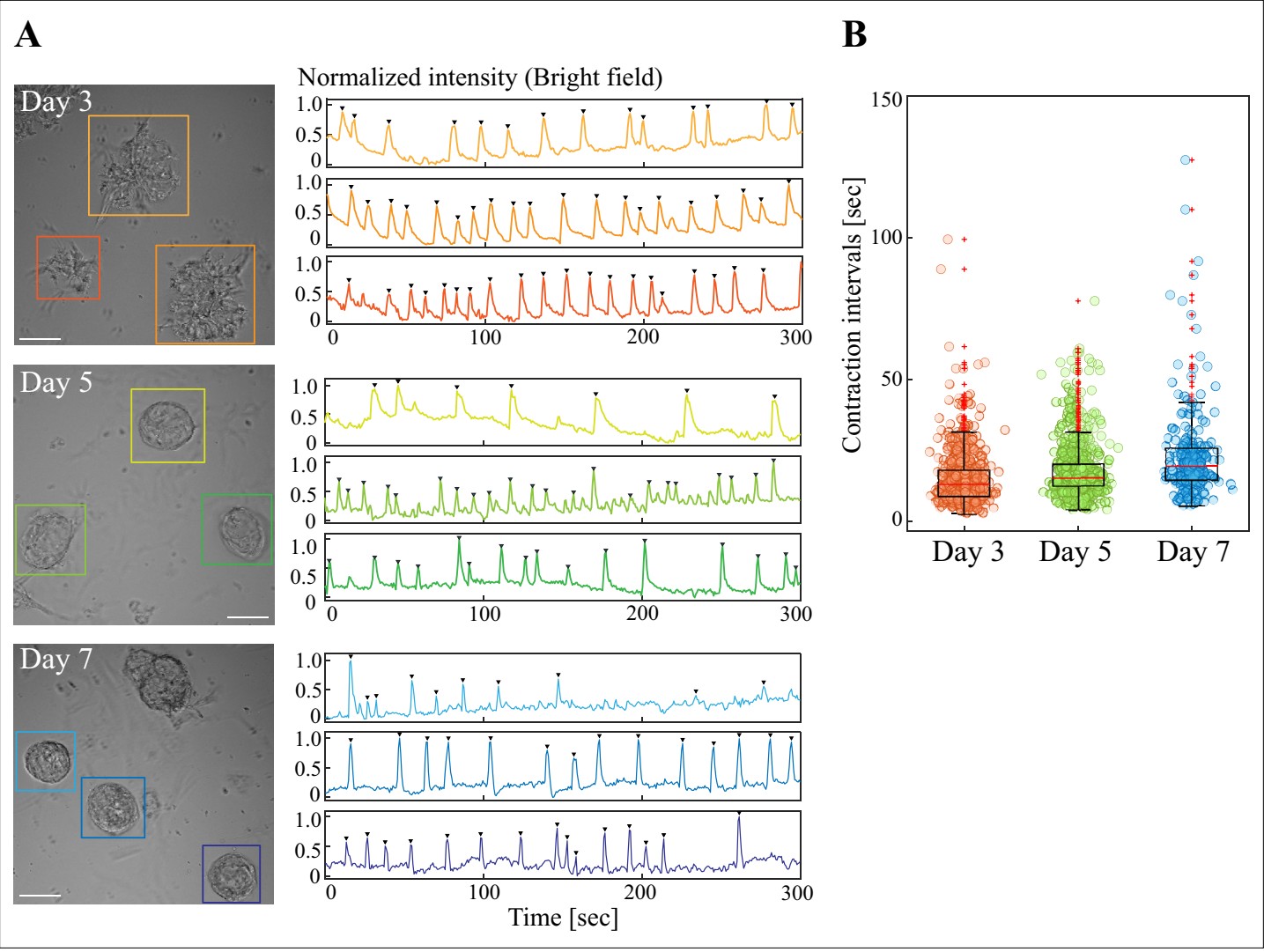

**Figure 2.** Spheroids formed in Neurobasal medium and Matrigel exhibited reiterated contractions. (**A**) Clusters/spheroids at days 3, 5, and 7 exhibited reiterative contractions. Graphs show normalized contraction intensities visualized using the Time Measurement function. Arrowheads indicate contraction peaks defined by a peak prominence > 0.25 and a peak width ≤ 10 seconds. (**B**) Contraction intervals in clusters/spheroids from day 3 to day 7. Each dot represents a single contraction interval. Median values: day 3, 13.3; day 5, 15.4; day 7, 19.6. Sample sizes: day 3, n=35, peak count=646; day 5, n=48, peak count=748; day 7, n=21, peak count=250. Scale bars: 50 μm (**A**).

The online version of this article includes the following video and source data for figure 2:

**Source data 1.** Time lapse data for *Figure 2* and *Figure 2—video 1*.

**Figure 2—video 1.** Periodic contractions of the cluster/spheroid on days 3, 5, and 7.

https://elifesciences.org/articles/97860/figures#fig2video1

Time-lapse images were taken with 700 ms intervals for 5 min. This video corresponds to *Figure 2A*. Scale bars: 50 μm.

*1A and B*). Indeed, when tetrodotoxin was added to culture medium, contraction intervals of organoids were comparable to those of control (before the administration) (*Figure 3—figure supplement 1C*), confirming little contribution by ENS. We counted cell numbers of each cell type per organoid and found that, contrasting with the intact gut, the proportion of SMCs was smaller than that of ICCs (*Figure 3—figure supplement 1B*), possibly due to a cell-type-specific loss during culture preparation.

Since ICCs and SMCs were spatially segregated in the sphenoid, we stained with anti N-cadherin antibody (E-cadherin is positive solely in the mucosa/endoderm; *Graham et al., 2017*; *Grosse et al., 2011*). N-cadherin signals were seen in the internal ICCs, and not in the peripheral SMCs (*Figure 3D*). Thus, it is likely that N-cadherin plays a role in the segregation.

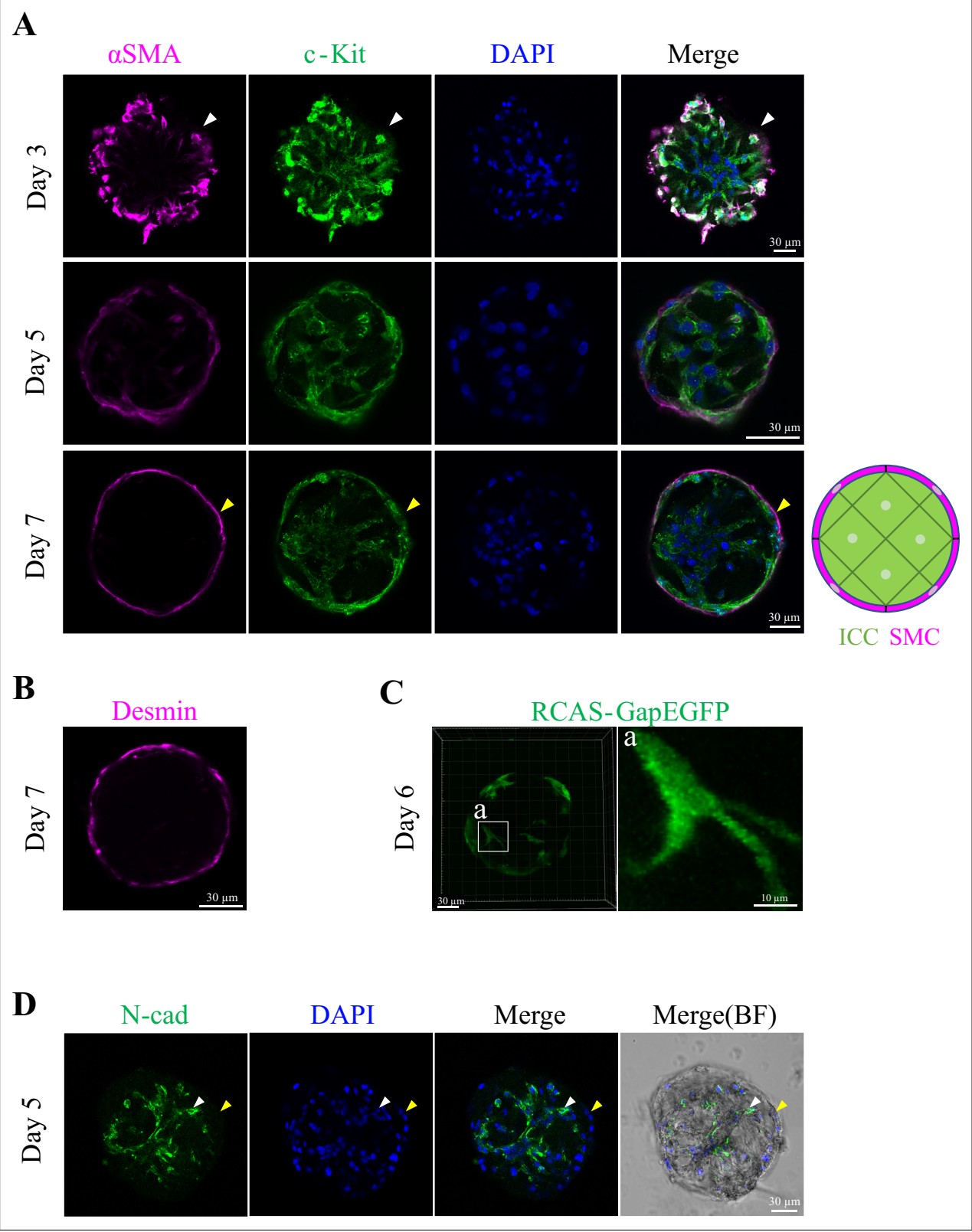

**Figure 3.** Clusters/spheroids are composed of internally located ICCs and peripherally located SMCs. (**A**) Co-staining with anti-c-Kit and anti-αSMA antibodies. White arrowheads indicate co-expression of c-Kit and αSMA at day 3. Yellow arrowheads indicate cells expressing αSMA but not c-Kit at day 7. A schematic diagram illustrates the spatial arrangement of cells within a day 7 spheroid (green: ICCs; magenta: SMCs). (**B**) Staining with anti-Desmin antibody, a marker for mature smooth muscle cells. (**C**) Cell morphology within the spheroid at day 6 visualized by RCAS-gapEGFP expression. (**D**)

*Figure 3 continued on next page*

*Figure 3 continued*

Staining of day 5 spheroids with anti-N-cadherin antibody. A white arrowhead indicates N-cadherin-positive cells, while a yellow arrowhead indicates N-cadherin–negative cells in the outer region of the spheroid. Scale bars: 30 μm (**A–D**), 10 μm (inset a in **C**).

The online version of this article includes the following source data and figure supplement(s) for figure 3:

**Figure supplement 1.** Neuronal markers and TTX response at day 7 organoids.

**Figure supplement 1—source data 1.** Cell counts and time-lapse data for *Figure 3—figure supplement 1*.

In summary, the hindgut-derived spheroid displays three prominent characteristics: (1) a dominant occupation by ICCs and SMCs with negligible contribution by ENS, (2) self-organization ability of internal ICCs encapsulated by a thin layer of SMCs, (3) recurrent and stable contractions. Based on these characteristics, we designated this spheroid as 'gut contractile organoid'.

## Contraction-associated intracellular Ca²⁺ transients were coordinated between ICCs and SMCs in the gut contractile organoid

It has been reported that $Ca^{2+}$ dynamics are important for pacemaker activity in ICCs and contractions of SMCs. $Ca^{2+}$ flows into ICCs via voltage-dependent $Ca^{2+}$ channels and propagates to SMCs, causing the gut muscle contractions (*Baker et al., 2021*). We therefore investigated the $Ca^{2+}$ dynamics in our gut contractile organoids. Organoid-forming cells were infected with a RCAS vector encoding GCaMP6s, a $Ca^{2+}$ indicator that emits EGFP fluorescence in response to $Ca^{2+}$ influx, and mRuby3 as a reporter (*Figure 4A*). GCaMP6s-organoids were subjected to time-lapse imaging analyses by confocal microscopy. As expected, the oscillatory rhythm of $Ca^{2+}$ transients as a whole organoid was highly concomitant with that of contractions (*Figure 4B*, *Figure 4—video 1*).

Taking advantage of the spatial segregation between ICCs (internal) and SMCs (peripheral; *Figure 3A and B*), we compared the $Ca^{2+}$ oscillatory rhythm among ICCs and SMCs (homotypically) and between ICC-SMC (heterotypically). We set up 3 regions of interest (ROIs) in the SMC layer with one ROI corresponding to one cell and captured the $Ca^{2+}$ transients. The three ROIs exhibited a synchronous pattern of $Ca^{2+}$ oscillations (*Figure 4C*, SMCs). Similarly, $Ca^{2+}$ oscillations in three ROIs in the central region (ICCs) were synchronous (*Figure 4C*, ICCs). These data highlight active communications taking place intercellularly within SMCs and ICCs, respectively. We further compared $Ca^{2+}$ oscillations between ICC-SMC by setting up one ROI in each of ICC and SMC. Again, the $Ca^{2+}$ rhythm was synchronized between these heterotypic cell types (*Figure 4C*, ICC/SMC). With a deeper scrutinization, a peak of $Ca^{2+}$ transient in ICC preceded that in SMC with a time lag (also called latency) of 700 ms on average (*Figure 4C*, ICC/SMC; *Figure 4D*, 104 peaks for 14 organoids), suggestive of a signal propagation from ICC to SMC consistent with previous reports (*Baker et al., 2021*).

## Gap junctions play a role in ICC-to-SMC signaling

Although a series of gut motility studies have proposed an importance of gap junctions, rigid evidence has been limited due to a lack of experimental model. Our gut contractile organoid should prove useful for clarifying the roles of gap junctions in the synchronous motility, since intercellular synchronization was observed between/among identifiable cells (ICCs and SMCs) in live as shown above. We performed pharmacological assessments using gap junction inhibitors including carbenoxolone (CBX) and 18β-glycyrrhetinic acid (18β-GA; *Chevalier, 2018*; *Takeda et al., 2005*), two of the most widely used pharmacological inhibitors for gap junction studies.

Following inhibitor administration into day 7 culture medium, organoids were allowed to rest for 30 min to exclude possible effects by the administration, for example, turbulence of the medium. Contrary to our expectation, none of the 100 μM concentrations of CBX or 18β-GA showed detectable effects on the recurring contractions of an organoid with frequency/interval comparable to the control (*Figure 5—figure supplement 1A and B*, *Figure 5—video 2*). These undetectable effects were not due to insufficient penetrance of the drugs into the organoid, since the drug administration at day 3, when organoid-forming cells were still at 2D in culture, also yielded undetectable effects assessed at day 7 (*Figure 5—figure supplement 1C*). The inhibiting activity of the drugs used here was verified using embryonic heart cultured cells (*Figure 5—figure supplement 1D*).

Such unchanged patterns were also observed for the synchronization/coordination of $Ca^{2+}$ transients either in ICCs or SMCs in experiments conducted in a way similar to *Figure 4* (*Figure 5A and B*,

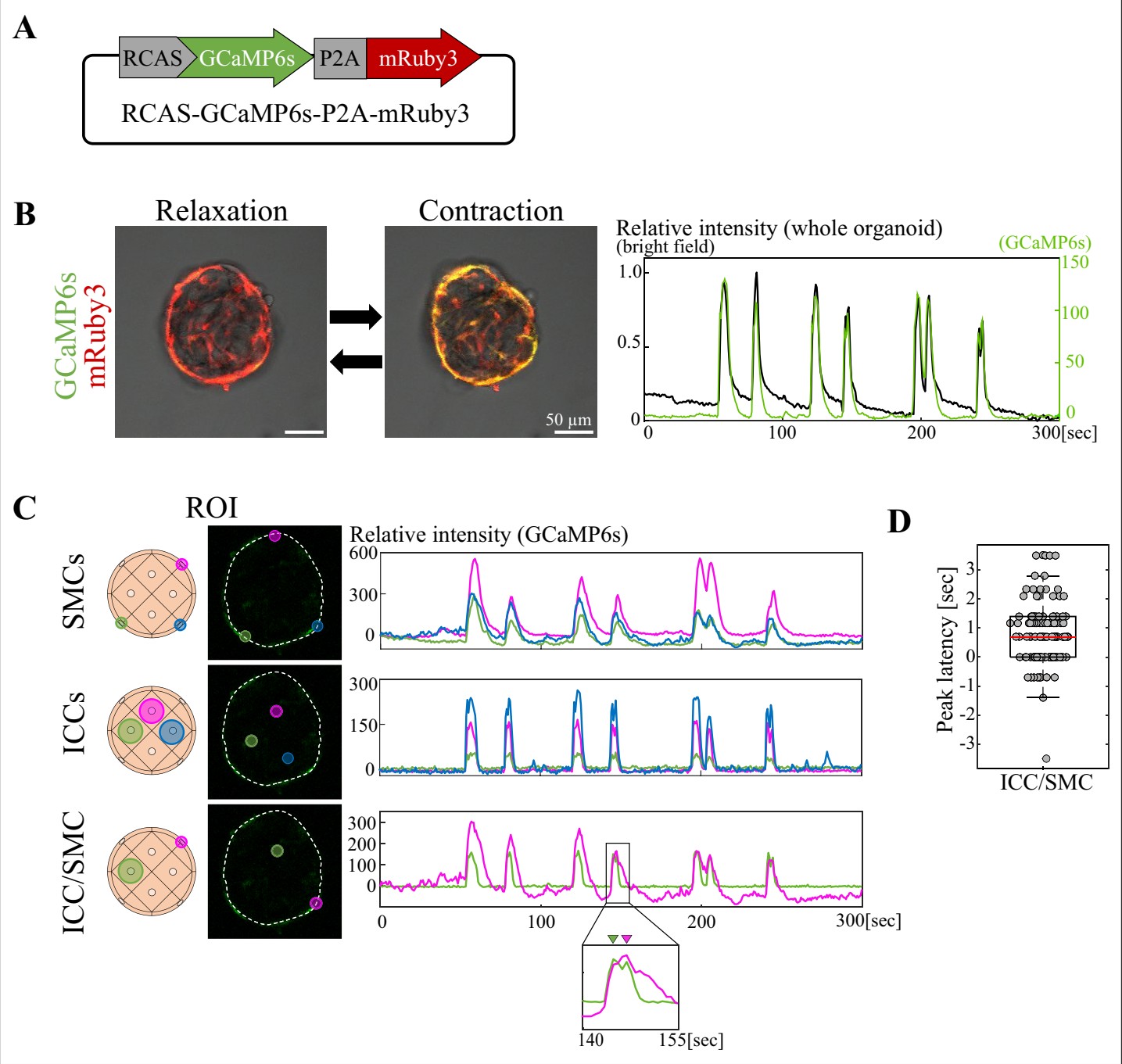

**Figure 4.** Ca²⁺ imaging of the gut contractile organoid revealed intercellular synchronization. (**A**) RCAS-GCaMP6s-P2A-mRuby3 plasmid. (**B**) Ca²⁺ imaging of gut contractile organoid during relaxation and contraction. Ca²⁺ dynamics (green) and normalized values of contraction (gray) of gut contractile organoid. (**C**) Simultaneous measurement of intercellular Ca²⁺ dynamics between ICC-ICC, SMC-SMC, or ICC-SMC. Three or two ROIs in Ca²⁺ signal-positive cells were set in a single organoid. Graphs show Ca²⁺ dynamics in the ROIs. Magnified view shows that a peak of Ca²⁺ signal in ICC (green) preceded that in SMC (magenta). (**D**) Peak latency in ICC/SMC. Median value = 0.7; Sample size: n = 14, peak count = 104. Scale bars: 50 μm (**B**).

The online version of this article includes the following video and source data for figure 4:

**Source data 1.** Time lapse data for *Figure 4* and *Figure 4—video 1*.

**Figure 4—video 1.** Ca²⁺ dynamics in day 7 gut contractile organoid are concomitant with its contractions.

https://elifesciences.org/articles/97860/figures#fig4video1

Time-lapse images were obtained with 700 ms intervals for 5 min. This video corresponds to *Figure 4*. Scale bar: 50 μm.

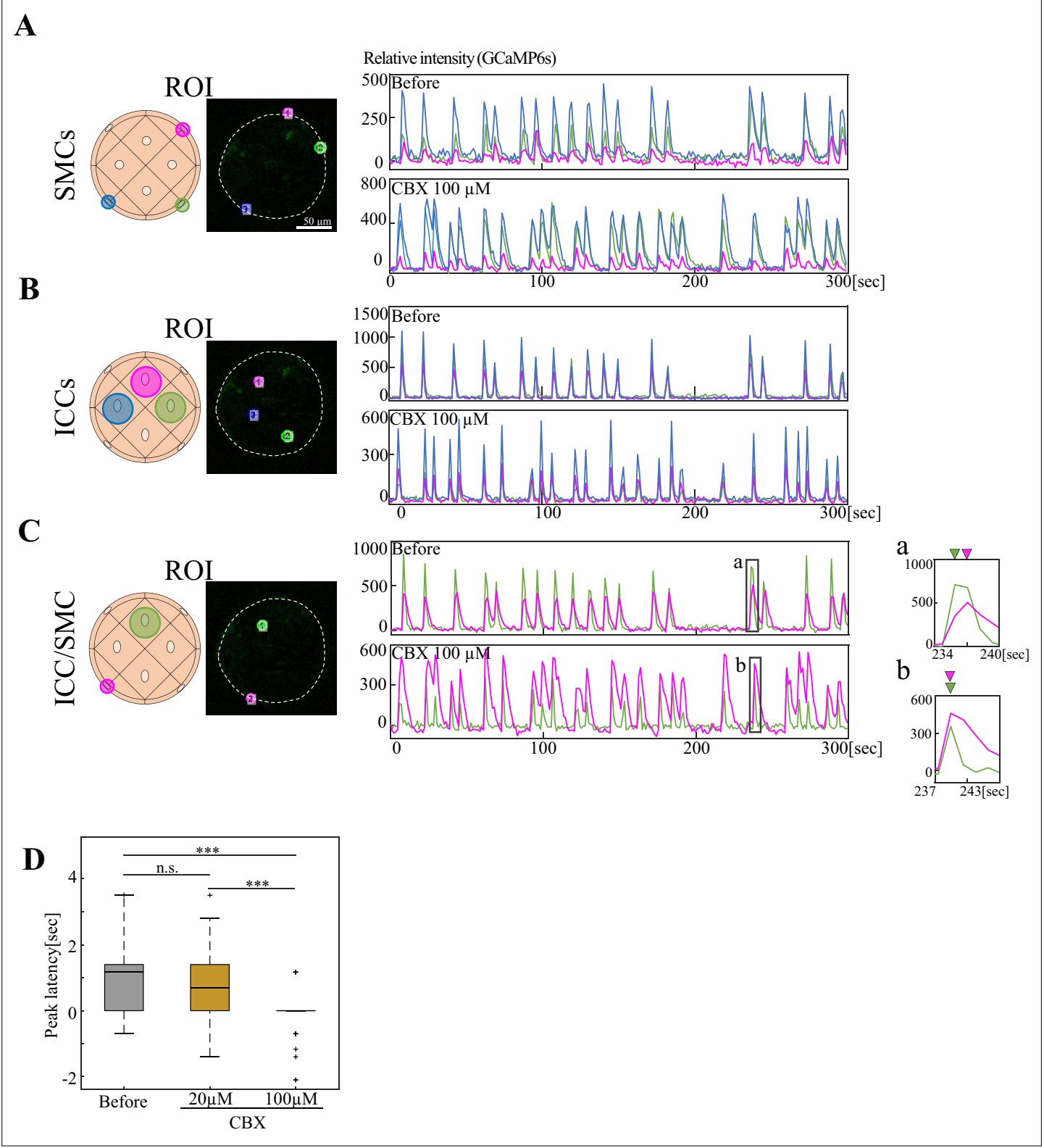

**Figure 5.** Gap junction inhibitor exerted limited effects on the synchronization of Ca²⁺ dynamics. Ca²⁺ synchronization among two or three ROIs in GCaMP6s-expressing organoids was evaluated before and after treatment with 100 µM CBX. The synchronization was unaffected between (**A**) SMC-SMC and (**B**) ICC-ICC, but was partially affected between (**C**) ICC–SMC. Magnified views (**a, b**) highlight that the preceding Ca²⁺ peak in ICC (green) observed before CBX treatment (**a**) was abolished after treatment (**b**). (**D**) Peak latency between ICC and SMC. Median values: Before, 1.167; CBX

*Figure 5 continued on next page*

*Figure 5 continued*

20 µM, 0.701; CBX 100 µM, 0.0. Statistical significance was assessed using Welch's t-test. \*\*\**p<0.001*, n.s., p≥0.05 (p=0.27 for Before vs. CBX 20 µM). Sample sizes: Before: n = 10, peak count = 50; CBX 20 µM: n = 5, peak count = 34; 100 µM: n = 4, peak count = 51, Scale bar: 50 µm (**A**).

The online version of this article includes the following video, source data, and figure supplement(s) for figure 5:

**Source data 1.** Time lapse data for *Figure 5* and *Figure 5—video 1*.

**Figure supplement 1.** Effects of gap junction inhibitors on organoidal contractions and ICC-SMC latency.

**Figure supplement 1—source data 1.** Time lapse data for *Figure 5—figure supplement 1* and *Figure 5—video 2*.

**Figure 5—video 1.** $Ca^{2+}$ dynamics in a gut contractile organoid with CBX.

https://elifesciences.org/articles/97860/figures#fig5video1

Time-lapse images were obtained with 700 ms intervals for 5 min. This video corresponds to *Figure 5*. Scale bars: 50 µm.

**Figure 5—video 2.** Periodic contractions of a day 7 organoid with CBX.

https://elifesciences.org/articles/97860/figures#fig5video2

Time-lapse images were obtained with 700 ms intervals for 5 min. This video corresponds to *Figure 5—figure supplement 1*. Scale bars: 50 µm.

*Figure 5—video 1*). Same cells in an organoid were tracked for their $Ca^{2+}$ transients before and after the inhibitor administration. With a closer look, however, while overall synchronization was retained between ICCs and SMCs, the preceding peak of $Ca^{2+}$ transient in ICC was abolished (*Figure 5C and D*). Collectively, while the contribution by gap junctions to the periodic contraction and intercellular synchronization in the Day 7 organoid is relatively limited, the ICC-to-SMC signals require gap junction-mediated communications, at least partly.

## Blebbistatin and Nifedipine ceased organoidal contractions and oscillatory patterns of $Ca^{2+}$ transients

Toward searching for factors that regulate the coordination between/among ICCs and SMCs, we tested blebbistatin, a specific inhibitor of myosin II, which was expected to cease organoidal contractions. Experimental procedures were similar to those for gap junction inhibitors. We found that blebbistatin ceased periodic contractions of organoids in a concentration-dependent manner: while intervals were shorter with smaller amplitude at 5 µM, contractions were ceased completely at 10 µM (*Figure 6A and B*, *Figure 6—video 1*). The 10 µM-treated specimens resumed contractions following medium washout, showing that the organoids were alive (*Figure 6B*).

We examined $Ca^{2+}$ transients in these contraction-inhibited organoids. Markedly, periodic $Ca^{2+}$ transients were extinguished not only in SMCs but also in ICCs, yielding no/little synchronous $Ca^{2+}$ patterns among and between ICCs and SMCs (*Figure 6C–F*). Although a possible direct inhibition of non-muscle myosin II in ICCs cannot be excluded, these findings raised an interesting possibility that the contractility feeds back to ICCs to generate/maintain their periodic rhythm.

This notion was further corroborated by similar experiments using Nifedipine, a blocker of L-type $Ca^{2+}$ channel known to function in gut SMCs (*Chevalier et al., 2024*; *Der et al., 2000*; *Der-Silaphet et al., 1998*). Organoidal contractions were completely ceased by 1 µM Nifedipine and resumed after washout, at least partly (*Figure 7A and B*). In Nifedipine-treated organoids, $Ca^{2+}$ transients (GCaMP) in ICCs were erased, which resumed following washout (*Figure 7C*).

## Inter-organoidal coordination was mediated by SMCs

During analyses with our novel organoids, we noticed that they easily fuse to each other, suggesting that organoids grow by progressive fusion (*Figure 8A*, *Figure 8—video 1*). This also raised the possibility that the synchronous $Ca^{2+}$ transients among organoid-constituting cells (*Figure 4C*) might be a consequence of phasic coordination upon the fusion of multiple organoids that had shown different oscillatory phases. To test this possibility, we transferred two organoids in a petri dish and allowed them to fuse. Before the fusion, $Ca^{2+}$ oscillatory rhythm was indeed out of phase in the two organoids (*Figure 8B*, *Figure 8—video 2*). Markedly, upon fusion by 24 hr, their rhythm became in phase/synchronous (*Figure 8B*, *Figure 8—video 2*). Intriguingly, they underwent a 'pause' of oscillation during fusion (see Discussion).

Since we noticed that cellular protrusions were often observed around the time of organoidal fusion, we reasoned that these cellular processes would mediate the fusion and its subsequent

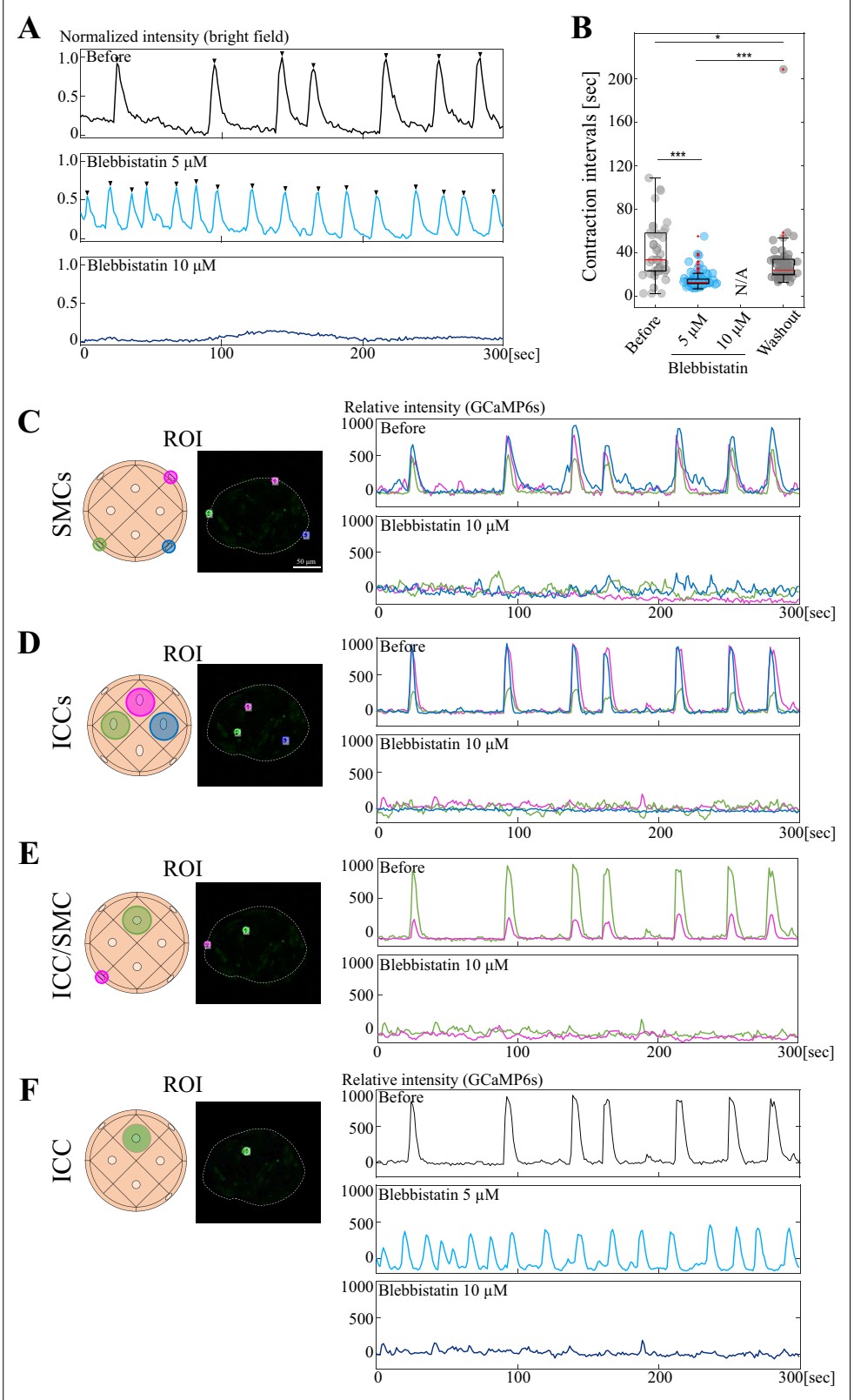

**Figure 6.** The organoidal contraction is important for Ca²⁺ dynamics in ICCs. GCaMP6s-expressing organoids were cultured with Blebbistatin. (**A**) Organoidal contractions were extinguished at 10 μM. (**B**) Contraction intervals before and after Blebbistatin, and upon washout. Median values: Before, 33.46; Blebbistatin 5 μM, 12.86; 10 μM, N/A; Washout, 23.95. Statistical significance was assessed using Welch's t-test. *\*\*\*p<0.001, \*p<0.05*. Sample size:

*Figure 6 continued on next page*

*Figure 6 continued*

n=3. (**C**–**E**) Comparison of Ca$^{2+}$ dynamics in SMC-SMC, ICC-ICC, and ICC-SMC. Three or two ROIs were assessed before and after administrations of 10 µM Blebbistatin. (**F**) Ca$^{2+}$ transients in a single ICC at 0 µM (before), 5 µM, and 10 µM. Scale bar: 50 µm (**B**).

The online version of this article includes the following video and source data for figure 6:

**Source data 1.** Time-lapse data for *Figure 6* and *Figure 6—video 1*.

**Figure 6—video 1.** Ca$^{2+}$ dynamics in day 7 gut contractile organoid with Blebbistatin.

https://elifesciences.org/articles/97860/figures#fig6video1

Time-lapse images were obtained with 700 ms intervals for 5 min. This video corresponds to *Figure 6*. Scale bars: 50 µm.

synchronization of Ca$^{2+}$ transients (*Figure 8D*). To test this, we developed a three-well hydrogel with narrow channels connecting the wells (*Figure 9A*). Organoids were placed separately in each well, which prevented the fusion of organoidal bodies but allowed extension of cellular processes through the narrow channels and contact with each other (*Figure 9B*). After 72 hr of placement of organoids, cellular processes as well as several cell bodies were present within the channel, and Ca$^{2+}$ transients became synchronized among the three organoids (*Figure 9C*, *Figure 9—video 1*). However, with careful examination, we also noticed that organoid-derived cells crawled out from the wells to cover the top surface of the hydrogel, connecting the three organoids (*Figure 9D*). Thus, another possibility was raised that these crawled-out cells would mediate the inter-organoidal synchronization.

To test this possibility, we prepared a similar three-well hydrogel, but in this case, the three wells were disconnected (no channels) so that organoid-derived cellular processes were not able to connect each other (*Figure 9E*). By 3 days of culture, the top surface of the hydrogel was indeed covered by cells in a similar way to *Figure 9D*. These cells were positive for αSMA but negative for c-Kit (*Figure 9F*), showing that they were SMCs that were somehow detached and crawled out from the peripheral layer of their 'host' organoids. Importantly, coinciding with the top coverage by the SMCs, the three organoids in the disconnected wells displayed synchronized Ca$^{2+}$ transients, highlighting the role of SMCs in mediating coordination between organoids (*Figure 9G*, *Figure 9—video 2*). Gap junction inhibitor yielded no/little effects on the Ca$^{2+}$ transient coordination (*Figure 9H*, *Figure 9—video 3*).

## Discussion

We have developed a novel gut contractile organoid, which displays several unique characteristics: (1) it undergoes recurrent contractions, (2) differentiation states of ICCs (c-Kit$^+$/αSMA$^-$) and SMCs (c-Kit$^-$/αSMA$^+$) are maintained at least until day 7 in the organoid, (3) the organoid is composed essentially of two types of cells, ICCs and SMCs, with few ENS cells, if any, (4) ICCs (internal) and SMCs (peripheral) can be distinguished for their localization in a living organoid, allowing (5) GCaMP-visualization of Ca$^{2+}$ transients and assessments of cell interactions between and among ICCs and SMCs. These characteristics circumvent, at least partly, obstacles that have hampered analyses in the research of gut peristalsis, such as unstable differentiation state of ICCs and SMCs in cultures, and difficulties in identifying these cells in living preparations. In studies of gut movements, how ICCs generate/maintain their periodic rhythm and how ICCs and SMCs interact with each other have been long-standing questions, and our organoids offer powerful advantages to address these fundamental questions and to understand the cellular mechanisms underlying the gut contractions/peristaltic motility at least in the embryonic gut.

Contrasting with many cases in organoid studies that aim at a *maximum* recapitulation of the intact organ, our gut contractile organoid is composed of a (probably) *minimum* number of cell types that suffice the generation and/or maintenance of rhythmic contractions, allowing high-resolution analyses at the cellular level. For example, adding ENS components to our organoid would allow the clarification of the role of ENS in the gut contraction. While the majority of internal cells in the organoid are ICCs (c-Kit$^+$/-αSMA$^-$), the possibility cannot be excluded that platelet-derived growth factor receptor-α positive (PDFGRα$^+$/c-Kit$^-$/αSMA$^-$) cells, another type of interstitial cells (fibroblast-like cell) known

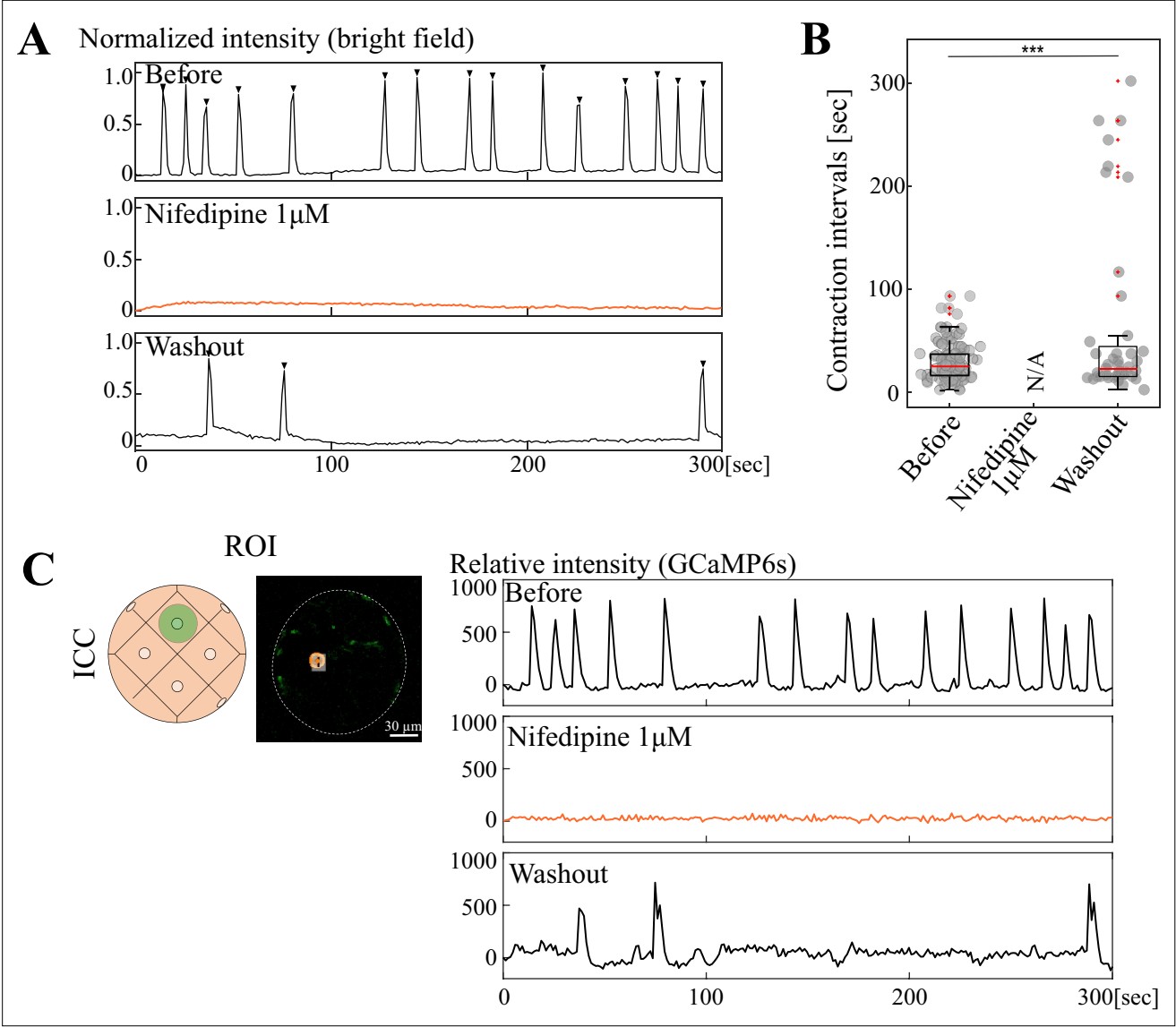

**Figure 7.** Nifedipine ceased organoidal contractions and Ca²⁺ activities in ICCs. Similar experiment to *Figure 6*. (**A**) Organoidal contractions were extinguished at 1 µM. (**B**) Contraction intervals before and after administration, and upon washout. Median values: Before, 25.7; Nifedipine 1 µM, N/A; Washout, 22.2. Statistical significance was assessed using Welch's t-test. **p<0.01. Sample size: n = 5. Peak counts: Before, 101; Nifedipine 1 µM, N/A; Washout, 46. (**C**) Ca²⁺ transients in a single ICC before and after administration of 1 µM nifedipine, and following washout. Scale bar: 30 µm.

The online version of this article includes the following source data for figure 7:

**Source data 1.** Time lapse data for *Figure 7*.

to mediate neural activity to SMCs in the mouse gut (*Sanders et al., 2024*; *Sanders et al., 2016*), are included in our organoid. An available antibody against the chicken PDFGRα protein is awaited.

## Coordinated Ca²⁺ transients in ICC/SMC populations in the gut contractile organoid

Measurement and quantification analyses with ICCs and SMCs that are identifiable in the living organoid revealed exquisite coordination of Ca²⁺ transients/oscillation homotypically in both ICC-ICC and SMC-SMC combinations, and heterotypically between SMC-ICC (*Figure 4C*). Notably, a peak of Ca²⁺ transient in ICCs precedes that of SMCs with a time lag (also called latency) of 700 msec, implying a signaling from ICC to SMC. These observations are consistent with previous studies using ICC- and SMC-specific transgenic mice that expressed GCaMP and RCaMP, respectively (*Baker et al., 2021*).

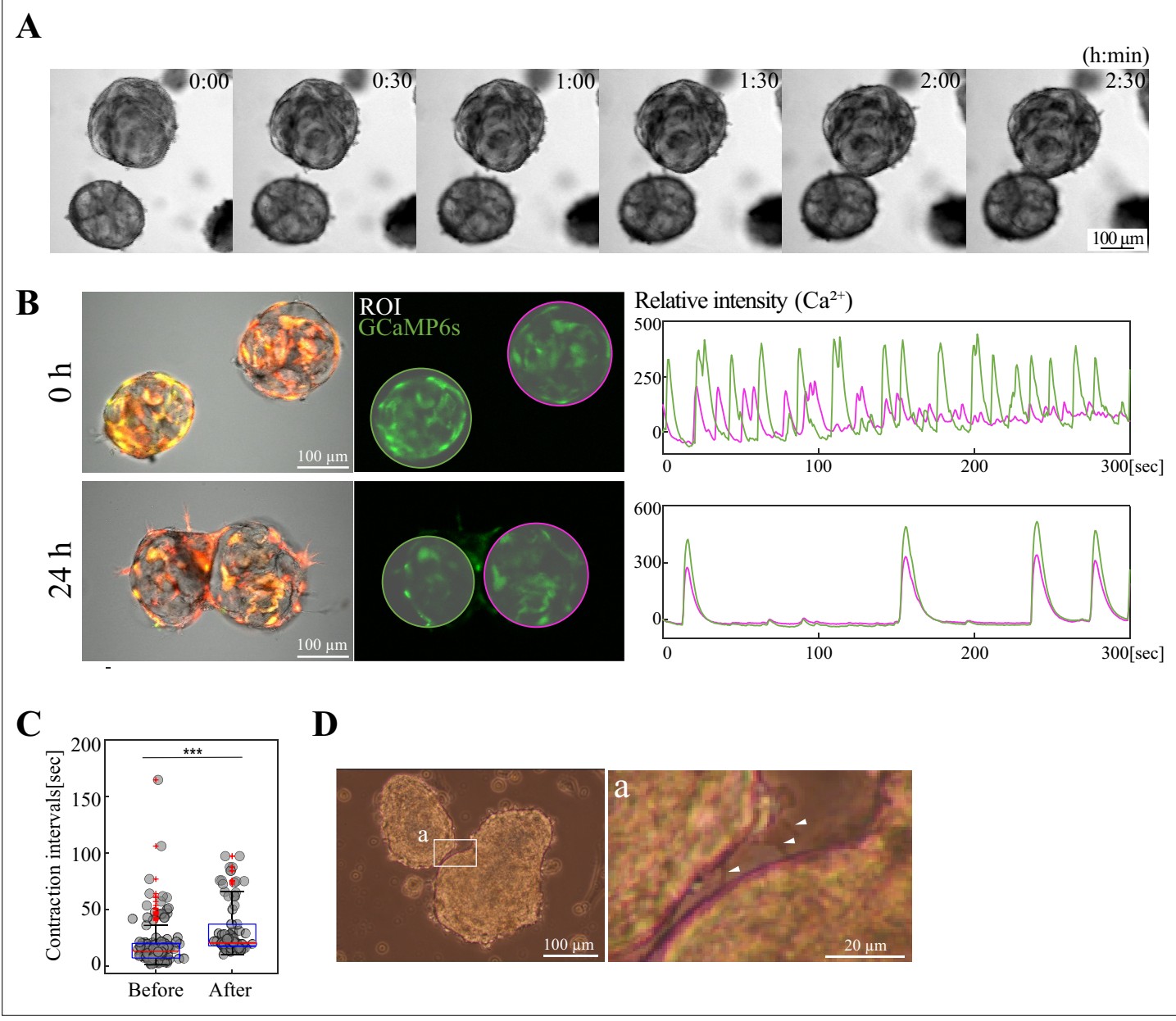

**Figure 8.** Ca²⁺ transients in multiple organoids undergo synchronization upon organoidal fusion. (**A**) Time-lapse imaging of organoidal fusion. (**B**) When two organoids that originally displayed independent Ca²⁺ rhythm fused to each other, their rhythm became synchronized after fusion (24 h). (**C**) Contraction intervals before and after fusion. Median values: before fusion, 13.65; after fusion, 21.04. Statistical significance was assessed using Welch's t-test. ***$p<0.001$. Sample size: n = 3 pairs. Peak counts: before, 138; after, 90. (**D**) Cellular protrusions between two neighboring organoids. White arrowheads show three protrusions from the left organoid. Scale bars: 100 μm (**A, B, D**), 20 μm (inset (**a**) in (**D**)).

The online version of this article includes the following video and source data for figure 8:

**Source data 1.** Time lapse data for *Figure 8* and *Figure 8—videos 1; 2*.

**Figure 8—video 1.** Live imaging during fusion of multiple organoids.

https://elifesciences.org/articles/97860/figures#fig8video1

Time-lapse images were obtained every 10 min for 4 hr. This video corresponds to *Figure 8A*. Scale bar: 50 μm.

**Figure 8—video 2.** Ca²⁺ transients in two gut contractile organoids before and after fusion.

https://elifesciences.org/articles/97860/figures#fig8video2

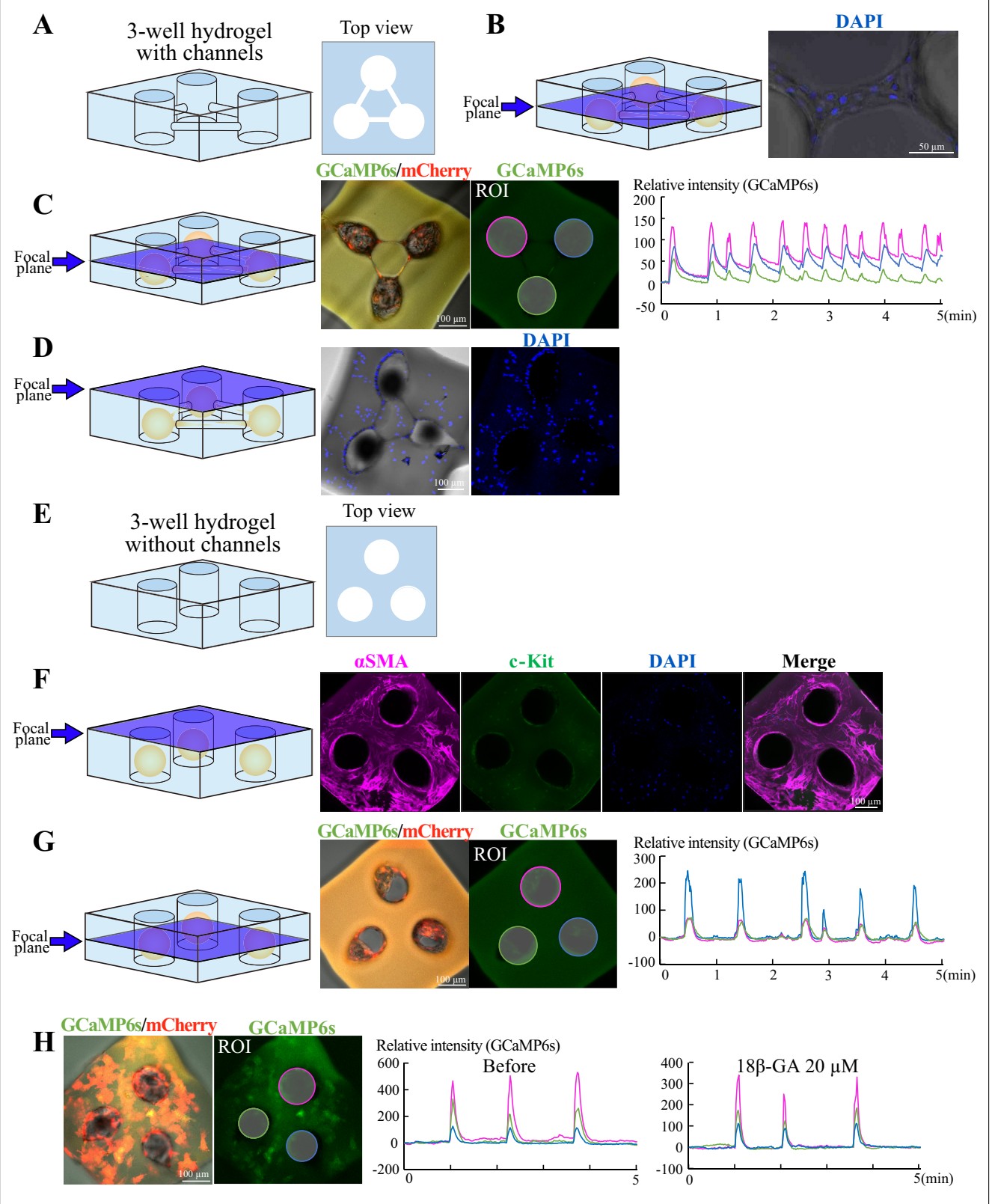

**Figure 9.** Smooth muscle cells mediate Ca²⁺ synchronization between organoids. (**A**) Diagram of a three-well hydrogel in which one organoid was placed per well. The three wells were connected with narrow channels, and this gel mold does not allow organoidal bodies to fuse to each other, but allows them to extend/migrate protrusions/cells through the channel (**B**). (**C**) After 3 days, the three organoids displayed synchronization of Ca²⁺ dynamics. (**D**) Some organoid-derived cells crawled out from the wells and covered the top surface of the hydrogel, resulting in bridging the three

*Figure 9 continued on next page*

*Figure 9 continued*

unfused organoids. (**E**) Diagram of a three-well hydrogel *without* channels. (**F**) The surface-covering cells were identified as SMCs (αSMA-positive, c-Kit-negative). (**G**) In the hydrogel without channels but with surface-covered SMCs, $Ca^{2+}$ dynamics in the three organoids were synchronized. (**H**) The $Ca^{2+}$ synchronization shown in (**G**) was not altered by treatment with 18β-GA. Blue planes indicate focal planes. Scale bars: 50 µm (**B**); 100 µm (**C, D, F, G, H**).

The online version of this article includes the following video and source data for figure 9:

**Source data 1.** Time lapse data for *Figure 9* and *Figure 9—video 1*, *Figure 9—video 2* and *Figure 9—video 3*.

**Figure 9—video 1.** $Ca^{2+}$ transients in three gut contractile organoids in three-well hydrogel with channels.

https://elifesciences.org/articles/97860/figures#fig9video1

Time-lapse images were obtained with 700 ms intervals for 5 min. This video corresponds to *Figure 9C*. Scale bar: 100 µm.

**Figure 9—video 2.** $Ca^{2+}$ transients in three gut contractile organoids in three-well hydrogel without channels.

https://elifesciences.org/articles/97860/figures#fig9video2

Time-lapse images were obtained with 450 ms intervals for 5 min. This video corresponds to *Figure 9G*. Scale bar: 100 µm.

**Figure 9—video 3.** Similar assay to *Figure 9—video 2* with 18β-GA administration.

https://elifesciences.org/articles/97860/figures#fig9video3

Time-lapse images were obtained with 700 ms intervals for 5 min. This video corresponds to *Figure 9H*. Scale bar: 100 µm.

In that study, the authors assessed $Ca^{2+}$ transients in submucosal ICCs (ICC-SM) and compared them with those in their adjacent circular muscles and showed that the rise of GCaMP signal (ICC) preceded that of RCaMP (SMC) with a latency of 56+/-14 ms. With these observations, they concluded that ICC-SM sends signals to its adjacent SMC. In our study, time-lapse imaging was mostly performed with 700 ms intervals. Further studies with shorter intervals are awaited to know whether the latency time would be shorter than 700 ms on average in our organoids.

## Contribution of gap junction to ICC-to-SMC signaling

Effects by the block of gap junction by CBX or 18β-GA were relatively limited in our organoid assays: organoidal contraction rhythm and synchronous patterns of $Ca^{2+}$ transients remained unchanged between SMC-SMC and ICC-ICC. In contrast, signaling from ICCs to SMCs was affected, in which the preceding peak of $Ca^{2+}$ transient in ICCs was abolished. This effect was seen by CBX but not by 18β-GA (*Figure 5D*, *Figure 5—figure supplement 1E*). Further studies are required to clarify which connexin(s) play a role in the generation of latency. The contribution of gap junctions to the ICC-to-SMC signaling was previously reported in mouse gut acting from intramuscular ICCs (ICC-IM) to their adjacent circular smooth muscles. However, interpretation of the role of gap junction in ICC-SMC interactions, in general, has been under big debate. Some studies reported that CBX or 18β-GA failed to inhibit these interactions or peristaltic motilities (*Komuro et al., 1996*; *Rohr et al., 1998*; *Schultz et al., 2003*), or that electron microscopy did not detect structures of gap junction in longitudinal muscles (*Cousins et al., 2003*; *Daniel and Wang, 1999*; *Gabella and Blundell, 1981*). The contribution of gap junctions to the gut motility appears to be highly variable in different regions of the gastrointestinal tract, for example, stomach versus colon (*Iino et al., 2007*; *Yang et al., 2012*). And a detection of gap junction structures or mRNA/protein of connexins does not necessarily mean that the gap junction is functional. A dominant negative form of a gap junction might be useful. Currently, ICC- or SMC-specific gene manipulations in our organoid are not available, and further studies are needed. One possibility is that while gap junctions are important for the onset of rhythm coordination, once the rhythm is established, other mechanisms might be employed, for example, mechanical feedback from muscles to ICCs (see below). A similar notion has recently been shown for coordinated $Ca^{2+}$ signaling in cardiac muscle cells (*Fukui et al., 2021*).

## Possible feedback from SMC's contractility to ICC's oscillatory rhythm

Blebbistatin extinguished the contraction of the organoids, which was concomitant with abrogation of $Ca^{2+}$ transients in ICCs (*Figure 6*). Since it has been reported that thick myosin fibers necessary for the contraction are found only in SMCs but not in ICCs (*Gherghiceanu and Popescu, 2005*; *Rumessen and Thuneberg, 1991*; *Rumessen and Thuneberg, 1996*; *Sun et al., 2006*), it is likely that ICCs do not have contractile ability. Our observations, therefore, raise the possibility that ICC's pacemaking activity requires mechanical feedback from contracting SMCs. This notion is also supported

by additional findings that Nifedipine, an L-type $Ca^{2+}$ channel blocker known to be expressed in SMCs (*Chevalier et al., 2024*; *Der et al., 2000*; *Der-Silaphet et al., 1998*), erases both organoidal contractions and $Ca^{2+}$ transients in ICCs (*Figure 7*). The possibility of the SMC-to-ICC signaling is further corroborated by other findings obtained in this study showing that SMCs mediate inter-organoidal rhythm coordination (*Figure 9*) as more discussed below.

It is of note that the reciprocal interactions between pace-making cells and their effectors have been reported in studies of neural circuit establishment. Spontaneous activities emerging in motor neurons during peristaltic locomotion of larvae in *Drosophila* are regulated by feedback from their governing muscle's contractions, so that motor neurons get organized to display coordinated and stable oscillatory activities (*Zeng et al., 2021*). Thus, it is tempting to speculate that during early gut peristalsis, ICCs that have initiated their spontaneous activities receive feedback from their governing SMCs to generate more stable coordination of pace-making activity among ICCs. Indeed, we have previously reported that in the very early embryonic gut, origins of peristaltic wave (OPWs) are randomly distributed along the gut axis, but these unstable OPWs later become confined to specific sites displaying more stable and coordinated pace-making patterns (*Shikaya et al., 2022*).

## Interactions between ICCs and SMCs in the gut contractile organoid

The organoid developed in this study was derived from the muscle layer (also called *tunica muscularis*) of chicken E15 hindgut devoid of mucosa and serosa, in which myenteric ICCs (ICC-MY), intramuscular ICCs (ICC-IM), and submucosal ICCs (ICC-SM) are localized in a way similar to those in mice, shown by staining with antibody against the chicken c-Kit protein (*Yagasaki et al., 2022*). In the current study, c-Kit antibody staining showed two types of cells in morphology in the gut contractile organoid: one is multipolar and N-cad-positive ICCs located centrally, and the other is ICCs that are thin in shape, N-cad negative, and lining beneath the most external layer of SMCs. Based on the knowledge obtained in studies with mammalian species that ICC-MY are multipolar whereas ICC-IM are bipolar and tightly associated with adjacent SMCs (*Huizinga et al., 2011*; *Iino and Horiguchi, 2006*), it is conceivable in our organoids that the central cells are ICC-MY, and the peripherally lining ones are ICC-IM. In addition, it has been reported in mice that signaling from ICC-IM to SMCs is gap junction-dependent. Collectively, our observations that gap junction-dependent signal found between ICC (central ICC) and SMC in the organoid (*Figure 5C and D*) could be interpreted as follows: ICC-MY (central) signals to ICC-IM (second-most peripheral), which in turn acts on the external SMCs mediated partly by gap junction. Such sequences of signaling (ICC-MY to ICC-IM to SMCs) have also been proposed in the intact gut in mammals, although rigid evidence has not been known. At present, direct comparison of $Ca^{2+}$ transients between the ICC-IM-like cells and SMCs in our organoid was technically very difficult since these two cells were both thin and tightly associated with each other.

## The gut contractile organoid provides a useful model and tool for studying phase coordination of oscillatory rhythm

To understand the gut peristaltic movements, which reiterate at specific sites along the gut axis (*Shikaya et al., 2022*), deciphering the mechanisms underlying the coordination/synchronization of oscillators among multiple cells is critical. Exploiting the finding obtained in the current study that multiple organoids easily fuse each other in vitro, we have found that two organoids with different oscillatory rhythms eventually coordinate their phases upon the fusion (*Figure 8*). This suggests the ability of ICCs to adjust their rhythm to their neighbors. It is tempting to speculate that during 'pausing time' of oscillation upon organoidal fusion (*Figure 8B*), they might communicate with each other to adjust to a unified rhythm.

During the identification of the rhythm-adjustment mediators using the three-well hydrogel (without connecting channels), SMCs that were unexpectedly crawled out from the organoids and covered the top surface of the hydrogel were able to mediate the rhythm coordination among organoids (*Figure 9*). Whether cellular thin protrusions connected with each other observed in the channel-connected three-well hydrogel mediate the coordination remains undetermined. Nevertheless, our findings have revealed a novel role of SMCs in mediating rhythm coordination, and as discussed above, these support the notion of the SMC-to-ICC signaling, which is unprecedented. It is unlikely that gap junctions play a major role in such signaling since gap junction inhibitors yielded no detectable effects in SMC-mediated organoidal phase synchronization (*Figure 9H*).

In summary, the gut contractile organoid developed in this study serves as a powerful model to study the establishment and maintenance of oscillatory rhythm (pace-making) and their coordination in the multicellular systems.

# Materials and methods

## Key resources table

| Reagent type (species) or resource | Designation | Source or reference | Identifiers | Additional information |
|---|---|---|---|---|
| Biological sample (*Gallus gallus*) | Embryonic hindgut | Yamagishi poultry farms (Wakayama, Japan) | | Freshly isolated from Gallus gallus |
| Biological sample (*Gallus gallus*) | Embryonic hindgut | Takeuchi Farm (Nara, Japan). | | Freshly isolated from Gallus gallus |
| Antibody | anti-c-Kit (Rabbit polyclonal) | Sigma Aldrich Japan; *Yagasaki et al., 2022* | | IF(1:300) |
| Antibody | anti-Tuj-1 (Mouse monoclonal) | RSD | MAB1195 RRID:AB_357520 | IF(1:500) |
| Antibody | anti-αSMA (Mouse monoclonal) | Sigma-Aldrich | A5228 RRID:AB_262054 | IF(1:400) |
| Antibody | anti-Desmin (Mouse monoclonal) | Novus Biologicals | NBP1-97707 RRID:AB_3243420 | IF(1:400) |
| Antibody | anti-chicken N-cadherin (Rat monoclonal) | TAKARA | M110 | IF(1:200) |
| Cell line (*Gallus gallus*) | DF-1 | ATCC | CRL-3586 RRID:CVCL_0570 | fibroblast cell line isolated from chicken embryo |
| Transfected construct | GCaMP6s-P2A-mRuby3 | Addgene | 112007 RRID:Addgene_112007 | |
| Chemical compound, drug | Carbenoxolone | nacalai tesque | 32775–51 | |
| Chemical compound, drug | 18beta-Glycyrrhetinic acid | abcam | ab142579 | |
| Chemical compound, drug | (-)-Blebbistatin | FUJIFILM Wako | 021–17041 | |
| Chemical compound, drug | Nifedipine | FUJIFILM Wako | 141–05783 | |
| Software, algorithm | MATLAB | MathWorks | findpeaks | |
| Other | Matrigel | Corning | 354248 | |
| Other | Neurobasal medium | Gibco | 21103–049 | |
| Other | 50× B-27 supplement | Gibco | 17504044 | |

## Chicken embryos

Fertilized chicken eggs were obtained from the Yamagishi poultry farms (Wakayama, Japan) and Takeuchi Farm (Nara, Japan). Embryos were staged according to embryonic days. All animal experiments were conducted with the ethical approval of Kyoto University (#202110).

## Culture preparation of hindgut-derived cells

A hindgut was dissected from E15 chicken embryos (*Figure 1—figure supplement 1A*) and cut into small pieces. After treating with 25 U dispase (Fujifilm Wako, 383–02281) /phosphate buffer saline (PBS: 0.14 M NaCl, 2.7 mM KCl, 10 mM $Na_2HPO_4$-$12H_2O$, 1.8 mM $KH_2PO_4$) at 38.5°C for 40 min, serosa and intestinal epithelium were removed using forceps. The muscle layer was minced into smaller pieces and treated with 0.2 mg/ml collagenase (Fujifilm Wako, 038–22361) and 0.25% trypsin/PBS at 37.0°C for 30 min. The reaction was stopped with 1% FBS/PBS followed by centrifugation at 800 rpm for 3 min. The pellet was resuspended and washed in PBS followed by centrifugation. They were suspended in culture medium, and $5.0 \times 10^5$ cells were plated on 14 mm diameter glass-bottom dishes (Matsunami, D11130H) which had been treated with undiluted Matrigel (Corning, 354248) at 38.5°C for 20 min. Poly-lysine and collagen-coated dishes were purchased (Matsunami, D11131H,

D11134H). D-MEM /Ham's F-12 (Wako, 048–29785), Ham's F-12 (Wako, 087–08335) and Neurobasal medium (Gibco, 21103–049) with 1×B-27 supplement (Gibco, 17504044) and 0.5 mM L-glutamine were tested. After seeding, time-lapse images were obtained with CM20 (Evident) under 5% $CO_2$ and 38.5°C with 2 hr intervals.

## Assessment of contraction in the spheroid/cluster and organoids

Time-lapse images were obtained using confocal microscopy (Nikon, A1R) under 5% $CO_2$ and 38.5°C. Region of interest (ROI) was set using the Time Measurement function in Nikon NIS Elements, and changes were detected in the mode 'stDev Intensity' and exported as individual CSV files. Bright-field changes were normalized to values between 0 and 1. For drug experiments, the values in the control group were used for normalization.

Using MATLAB (MathWorks), the changes were plotted, and peaks with a minimum prominence of 0.25 and a maximum width of 10 s were detected as contractions of spheroid/cluster or organoid using the findpeaks function. To determine the contraction frequency, we analyzed intervals between contractions and displayed them as box plots.

## Immunocytochemistry

An organoid was fixed in acetic acid/ethanol (1:5) for 10 min at room temperature (RT), and washed in PBS for 10 min at RT. The specimens were permeabilized in 0.1% Tween-20 in PBS for 10 min at RT, followed by washing in PBS for 10 min at RT. After blocking with 1% blocking reagent for 1 hr at RT, specimens were incubated overnight at 4°C with dilution of 1:300 anti-c-Kit (*Yagasaki et al., 2022*), 1:300 Tuj-1 (RSD, MAB1195), 1:400 anti-αSMA antibody (Sigma-Aldrich, A5228), anti-Desmin antibody (Novus Biologicals, NBP1-97707) and/or 1:200 anti-N-cadherin antibodies (TAKARA, M110) in 1% blocking reagent (Roche, 1096176)/PBS. Following three times washing in PBS for 10 min each at RT, specimens were incubated for 1.5 hr at RT with 1:500 anti-rabbit IgG(H+L)-Alexa 488-conjugated antibody (Donkey; Invitrogen, A21206), anti-mouse IgG$_{2a}$-Alexa 568-conjugated antibody (Goat; Invitrogen, A21134), anti-rat IgG (H+L)-Alexa 488-conjugated antibody (Goat; Invitrogen, A11006) and 1:2000 DAPI. After washing three times in PBS for 10 min at RT, fluorescent images were obtained using the Nikon A1R confocal microscope.

## Plasmids

pAAV-hSynapsin1-GCaMP6s-P2A-mRuby3 was purchased from Addgene (112007). Full-length cDNA of GCaMP6s was PCR-amplified:

> forward 5'- GCGTACCACTGTGGCATCGATGCCACCATGGGTTCTCA –3',
> reverse 5'- GCCCGTACATCGCATCGATTTACTTGTACAGCTCGT –3'.

The retroviral vector RCAS-EGFP was digested with ClaI to remove EGFP, into which a DNA fragment was inserted by In-Fusion HD Cloning Kit (TAKARA) to produce RCAS-GCaMP6s-P2A-mRuby3. RCAS-GapEGFP is as previously described (*Murai et al., 2015*).

## Preparation of retroviral vector particles

RCAS-GCaMP6s-P2A-mRuby3 was transfected into the chicken fibroblast line DF-1 cells (ATCC, CRL-3586), which were confirmed to be mycoplasma-negative, using Lipofectamine 2000 (Invitrogen). Transfected cells were cultured in a 10 cm culture dish until confluent. The supernatant of transfected DF1 was collected for viral precipitation, from which retroviral particles were prepared using Retro-Concentin Virus Precipitation Solution (SBI, RV100A-1). Since DF-1 cells were used solely for retrovirus production and not for any downstream experimental analysis, cell line authentication was not performed.

## Intracellular Ca²⁺ imaging in the gut contractile organoid

On day 2 or 3 of cell culture, a 10 mg/ml polybrene solution (final concentration: 4 μg/ml; Nacalai, 12996–81) and 20 μl of Opti-MEM containing the aforementioned viral particles were added to the culture medium to transfect GCaMP6s into organoid-forming cells. The culture medium was replaced with fresh medium on day 5, and $Ca^{2+}$ imaging was performed on day 7. Time-lapse images were acquired using a confocal microscope (Nikon, A1R) at intervals of either 700 or 450 ms. Fluorescence

intensity from each region of interest (ROI) was exported into an Excel file using the time-measurement function of the NIS-Elements software (Nikon). Fluorescence intensity traces were plotted using the Excel data, with the start of measurement set as time zero. For peak latency analysis, calcium transients in ICCs and SMCs were identified using the 'findpeaks' function in MATLAB (MathWorks). Temporal differences between peak signals in ICC and SMC were calculated to show the latency.

### Drug administration

Carbenoxolone (CBX; nacalai tesque, 32775–51) /$H_2O$, 18beta-Glycyrrhetinic acid (18β-GA; abcam, ab142579)/DMSO, (-)-Blebbistatin (FUJIFILM Wako, 021–17041) and Nifedipine (FUJIFILM Wako, 141–05783) were prepared. Time-lapse images were acquired for a single organoid before and after the drug administration. Following the drug addition, organoids were allowed to rest for 30 min to avoid possible effect of turbulence of the medium, and time-lapse images were taken for 5 min.

### Three-well hydrogel fabrication

The photoinitiator P2CK was synthesized as previously reported (*Li et al., 2013*). The target product was verified using proton NMR and Bruker's TopSpin software. Gelatin-Norbornene was synthesized based on a previous report (*Van Hoorick et al., 2018*). The final reaction mixture was transferred to a dialysis tube (5–6 kDa; Spectra Por, cat. no. 132680T) for dialysis against pure water at 40°C for 3 days. After dialysis, the solution's pH was carefully adjusted to 8.0.

A piece of three-well hydrogel was fabricated on a glass bottom dish (Matsunami) from 100 µL of a solution containing 20 wt % Gelatin-Norbornene, 2 mM of the photoinitiator P2CK, and 20 mM of crosslinker (DTT) (TCI, #D1071) using a two-photon microscope with controllable laser power in 3D space according to the voxel file input (Olympus, in-house customized). The voxel files specifying the 3D shape of the hydrogel were designed with Fusion 360 software (Autodesk). Organoids were transferred to the hydrogel using a glass capillary.

### Culture of embryonic heart-derived cells

A heart was dissected from an E10 chicken embryo and cut into small pieces. The tissue fragments were treated with 0.125% trypsin in PBS at 37.0°C for 30 min. The reaction was terminated by adding 10% FBS in D-MEM/Ham's F-12, followed by centrifugation at 800 rpm for 3 min. The pellet was resuspended in D-MEM/Ham's F-12 supplemented with 10% FBS, and the cells were plated in poly-L-lysine-coated glass-bottom dishes (Matsunami).

On day 1, time-lapse images were acquired using confocal microscopy under 5% $CO_2$ at 38.5°C, with 700 ms intervals. Cell contractions were detected using the 'stDev Intensity' mode in the Time Measurement function of Nikon NIS-Elements.

## Acknowledgements

We thank Dr Scott Gilbert for careful reading of the manuscript and discussion. We also thank the National BioResource Project (Chicken-Quail, Nagoya University) for their technical help. This work was supported by JSPS KAKENHI Grant Numbers; 23H04933, 20H03259, 20K20520, 20K21425 for YT, and 21K06198, 23H04702 for MI, and FY 2022 Kusunoki 125 of Kyoto University 125[th] Anniversary Fund for MI, and Ginpu Funds (Kyoto University) and incu-be fund (Leave a Nest Co., Ltd.) for RY. RY is an ex-fellow of JSPS.

## Additional information

### Funding

| Funder | Grant reference number | Author |
| --- | --- | --- |
| Japan Society for the Promotion of Science | KAKENHI 23H04933 | Yoshiko Takahashi |
| Japan Society for the Promotion of Science | KAKENHI 20H03259 | Yoshiko Takahashi |

| Funder | Grant reference number | Author |
|---|---|---|
| Japan Society for the Promotion of Science | KAKENHI 20K20520 | Yoshiko Takahashi |
| Japan Society for the Promotion of Science | KAKENHI 20K21425 | Yoshiko Takahashi |
| Japan Society for the Promotion of Science | KAKENHI 21K06198 | Masafumi Inaba |
| Japan Society for the Promotion of Science | KAKENHI 23H04702 | Masafumi Inaba |
| Kyoto University | Kusunoki 125 of Kyoto University 125th Anniversary Fund | Masafumi Inaba |
| Kyoto University | Ginpu Funds | Rei Yagasaki |
| Leave a Nest | incu-be fund | Rei Yagasaki |

The funders had no role in study design, data collection and interpretation, or the decision to submit the work for publication.

## Author contributions

Rei Yagasaki, Conceptualization, Resources, Data curation, Formal analysis, Validation, Investigation, Visualization, Methodology, Writing – original draft; Ryo Nakamura, Yuuki Shikaya, Data curation; Ryosuke Tadokoro, Data curation, Formal analysis, Investigation, Visualization, Methodology; Ruolin Hao, Zhe Wang, Methodology; Mototsugu Eiraku, Data curation, Methodology; Masafumi Inaba, Data curation, Investigation, Methodology; Yoshiko Takahashi, Conceptualization, Data curation, Supervision, Funding acquisition, Investigation, Methodology, Writing – original draft, Project administration, Writing – review and editing

## Author ORCIDs

Rei Yagasaki ⓘ https://orcid.org/0009-0005-7830-4391
Ryo Nakamura ⓘ https://orcid.org/0000-0002-0761-0704
Yuuki Shikaya ⓘ https://orcid.org/0009-0009-3544-9725
Ryosuke Tadokoro ⓘ https://orcid.org/0000-0001-7556-4263
Mototsugu Eiraku ⓘ https://orcid.org/0000-0002-9433-9085
Masafumi Inaba ⓘ https://orcid.org/0000-0001-6896-5819
Yoshiko Takahashi ⓘ https://orcid.org/0000-0002-1596-7527

## Ethics

All animal experiments were conducted with the ethical approval of Kyoto University (#202110). All of the chicken embryos were handled according to approved institutional animal care and use committee protocols (#202408) of Kyoto University.

Reviewer #1 (Public review): https://doi.org/10.7554/eLife.97860.3.sa1
Reviewer #2 (Public review): https://doi.org/10.7554/eLife.97860.3.sa2
Reviewer #3 (Public review): https://doi.org/10.7554/eLife.97860.3.sa3
Author response https://doi.org/10.7554/eLife.97860.3.sa4

# Additional files

## Supplementary files
MDAR checklist

## Data availability

All data generated or analysed during this study are included in the manuscript and supporting files; source data files have been provided for Figures 2, 4,5,6,7,8,9, Figure 3—figure supplement 1 and Figure 5—figure supplement 1.

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
