## [Editor Report · eLife Assessment]

This **valuable** study reports the development of a novel organoid system for studying the emergence of autorhythmic gut peristaltic contractions through the interaction between interstitial cells of Cajal and smooth muscle cells. The authors further utilized the system to provide **convincing** evidence for a previously unappreciated potential role for smooth muscle cells in regulating the firing rate of interstitial cells of Cajal. The work will be of interest to those studying development and physiology of the gut.

---

## [Referee Report · Reviewer #1 (Public review)]

Summary:

In this study, the authors developed an organoid system containing smooth muscle cells (SMCs) and interstitial cells of Cajal (ICCs; pacemaker cells), but few enteric neurons. This system generates rhythmic contractions similar to those observed in the developing gut. The stereotypical arrangement of SMCs and ICCs within the organoid allowed the authors to identify these cell types without the need for antibody staining. Leveraging this feature, they used calcium imaging and pharmacological approaches to investigate how calcium transients develop through interactions between the two cell types.

The authors first show that calcium transients are synchronized among ICC-ICC, SMC-SMC, and SMC-ICC pairs. They then used gap junction inhibitors to suggest that gap junctions are specifically involved in ICC-to-SMC signaling. Finally, they applied inhibitors of myosin II and L-type Ca²⁺ channels to demonstrate that SMC contraction is crucial for the generation of rhythmic activity in ICCs, suggesting the presence of SMC-to-ICC signaling. Additionally, they show that two organoids become synchronized upon fusion, with SMCs mediating this synchronization.

Strengths:

The organoid system provides a useful model for studying the specific roles of SMCs and ICCs in live samples.

Weaknesses:

Since all functional analyses were conducted pharmacologically in vitro, the findings need to be further validated through genetic approaches in vivo in future studies.

---

## [Referee Report · Reviewer #2 (Public review)]

Summary:

In this study, Yagasaki et al. describe an organoid system to study the interactions between smooth muscle cells (SMCs) and interstitial cells of Cajal (ICCs). While these interactions are essential for the control of rhythmic intestinal contractility (i.e., peristalsis), they are poorly understood, largely due to the complexity of and access to the in vivo environment and the inability to co-culture these cell types in vitro for long term under physiological conditions. The "gut contractile organoids" organoids described herein are reconstituted from stromal cells of the fetal chicken hindgut that rapidly reorganize into multilayered spheroids containing an outer layer of smooth muscle cells and an inner core of interstitial cells. The authors demonstrate that they contract cyclically and additionally use calcium imagining to show that these contractions occur concomitantly with calcium transients that initiate in the interstitial cell core and are synchronized within the organoid and between ICCs and SMCs. Furthermore, they use several pharmacological inhibitors to show that these contractions are dependent upon non-muscle myosin activity and, surprisingly, independent of gap junction activity. Finally, they develop a 3D hydrogel for the culturing of multiple organoids and found that they synchronize their contractile activities through interconnecting smooth muscle cells, suggesting that this model can be used to study the emergence of pacemaking activities. Overall, this study provides a relatively easy-to-establish organoid system that will be of use in studies examining the emergence of rhythmic peristaltic smooth muscle contractions and how these are regulated by interstitial cell interactions. However, further validation and quantification will be necessary to conclusively determine show the cellular composition of the organoids and how reproducible their behaviors are.

Strengths:

This work establishes a new self-organizing organoid system that can easily be generated from the muscle layers of the chick fetal hindgut to study the emergence of spontaneous smooth muscle cell contractility. A key strength of this approach is that the organoids seem to contain few cell types (though more validation is needed), namely smooth muscle cells (SMCs) and interstitial cells of Cajal (ICCs). These organoids are amenable to live imaging of calcium dynamics as well as pharmacological perturbations for functional assays, and since they are derived from developing tissues, the emergence of the interactions between cell types can be functionally studied. Thus, the gut contractile organoids represent a reductionist system to study the interactions between SMCs and ICCs in comparison to the more complex in vivo environment, which has made studying these interactions challenging.

Weaknesses:

The study lacks complementary in vivo experiments, but these will be exciting to follow up in future studies.

---

## [Referee Report · Reviewer #3 (Public review)]

Summary:

The paper presents a novel contractile gut organoid system that allows for in vitro studying of rudimentary peristaltic motions in embryonic tissues by facilitating GCaMP-live imaging of Ca2+dynamics, while highlighting the importance and sufficiency of ICC and SMC interactions in generating consistent contractions reminiscent of peristalsis. It also argues that ENS at later embryonic stages might not be necessary for coordination of peristalsis.

Strengths:

The manuscript by Yagasaki, Takahashi, and colleagues represents an exciting new addition to the toolkit available for studying fundamental questions in the development and physiology of the hindgut. The authors carefully lay out the protocol for generating contractile gut organoids from chick embryonic hindgut and perform a series of experiments that illustrate the broader utility of these organoids for studying the gut. This reviewer is highly supportive of the manuscript following highly responsive revisions in response to prior reviewer feedback.

---

## [Author Response]

The following is the authors’ response to the original reviews.

**eLife Assessment**
This valuable study reports the development of a novel organoid system for studying the emergence of autorhythmic gut peristaltic contractions through the interaction between interstitial cells of Cajal and smooth muscle cells. While the utility of the organoids for studying hindgut development is well illustrated by showing, for example, a previously unappreciated potential role for smooth muscle cells in regulating the firing rate of interstitial cells of Cajal, some of the functional analyses are incomplete. There are some concerns about the specificity and penetrance of perturbations and the reproducibility of the phenotypes. With these concerns properly addressed, this paper will be of interest to those studying the development and physiology of the gut.

We greatly appreciate constructive comments raised by the Editors and all the Reviewers. We have newly conducted pharmacological experiments using Nifedipine, a L-type Ca^2+^ blocker known to operate in smooth muscles (new Fig 7). The treatment abrogated not only the oscillation of SMCs but also that in ICCs, further corroborating our model that not only ICC-to-SMC interactions but also the reverse direction, namely SMC-to-ICC feedback signals, are operating to achieve coordinated/stable rhythm of gut contractile organoids.

Concerning the issues of the specificity and penetrance in pharmacological experiments with gap junction inhibitors, we have carefully re-examined effects by multiple blockers (CBX and 18b-GA) at different concentrations (new Fig 5D and Fig. S3B).We have newly found that: (1) the effects observed by CBX (100 µM) that the latency of Ca^2+^ peaks between ICCs (preceding) and SMCs (following) was abolished are not seen by 18b-GA at any concentrations including 100 µM, implying that the latency of Ca^2+^ peaks between these cells is governed by connexin(s) that are not inhibited by18bGA. Such difference in inhibiting effects by these two drugs were previously reported in multiple model systems including guts (Daniel et al., 2007; Parsons & Huizinga, 2015; Schultz et al., 2003).

Regarding the penetrance of the drugs, we have carried out earlier administration (Day 3) of the gap junction inhibitor, either CBX (100 µM) or 18b-GA (100 µM), in the course of organoidal formation in culture when cells are still at 2D to exclude a possible penetrance problem (new Fig. S3C). There treatments render no or little effects to the patterns of organoidal contractions in a way similar to the drug administration at Day 7. As already shown in the first version, CBX (100 µM) eliminates the latency of Ca^2+^ peaks, we believe that this drug successfully penetrates into the organoid and exerts its specific effects.

Unfortunately, due to very unstable condition in climate including extreme heat and sporadically occurring bird flu epidemic since the last summer in Japan, the poultry farm must have faced problems. In the course of revision experiments, we got in a serious trouble at multiple times with unhealthy eggs/embryos lasting from last summer until present. These unfortunate incidents did not allow us to engage in the revision experiments as fully as we originally planned. Nevertheless, we did our very best within a limited time fame, and we believe that the revised version is suitable as a final version of an eLife article.

**Public Reviews:**

**Reviewer #1 (Public Review):**
Summary:In this study, the authors developed an organoid system that contains smooth muscle cells (SMCs) and interstitial cells of Cajal (ICCs; pacemaker) but few enteric neurons, and generates rhythmic contractions as seen in the developing gut. The stereotypical arrangements of SMCs and ICCs in the organoid allowed the authors to identify these cell types in the organoid without antibody staining. The authors took advantage of this and used calcium imaging and pharmacology to study how calcium transients develop in this system through the interaction between the two types of cells. The authors first show that calcium transients are synchronized between ICC-ICC, SMC-SMC, and SMC-ICC. They then used gap junction inhibitors to suggest that gap junctions are specifically involved in ICC-to-SMC signaling. Finally, the authors used an inhibitor of myosin II to suggest that feedback from SMC contraction is crucial for the generation of rhythmic activities in ICCs. The authors also show that two organoids become synchronized as they fuse and SMCs mediate this synchronization.Strengths:The organoid system offers a useful model in which one can study the specific roles of SMCs and ICCs in live samples.

Thank you very much for the constructive comments.

Weaknesses:Since only one blocker each for gap junction and myosin II was used, the specificities of the effects were unclear.

We appreciate these comments. We have addressed those of “weaknesses” as described in “Responses to the eLife assessment” (please see above).

**Reviewer #2 (Public Review):**
Summary:In this study, Yagasaki et al. describe an organoid system to study the interactions between smooth muscle cells (SMCs) and interstitial cells of Cajal (ICCs). While these interactions are essential for the control of rhythmic intestinal contractility (i.e., peristalsis), they are poorly understood, largely due to the complexity of and access to the in vivo environment and the inability to co-culture these cell types in vitro for long term under physiological conditions. The "gut contractile organoids" organoids described herein are reconstituted from stromal cells of the fetal chicken hindgut that rapidly reorganize into multilayered spheroids containing an outer layer of smooth muscle cells and an inner core of interstitial cells. The authors demonstrate that they contract cyclically and additionally use calcium imagining to show that these contractions occur concomitantly with calcium transients that initiate in the interstitial cell core and are synchronized within the organoid and between ICCs and SMCs. Furthermore, they use several pharmacological inhibitors to show that these contractions are dependent upon non-muscle myosin activity and, surprisingly, independent of gap junction activity. Finally, they develop a 3D hydrogel for the culturing of multiple organoids and found that they synchronize their contractile activities through interconnecting smooth muscle cells, suggesting that this model can be used to study the emergence of pacemaking activities. Overall, this study provides a relatively easy-to-establish organoid system that will be of use in studies examining the emergence of rhythmic peristaltic smooth muscle contractions and how these are regulated by interstitial cell interactions. However, further validation and quantification will be necessary to conclusively determine show the cellular composition of the organoids and how reproducible their behaviors are.Strengths:This work establishes a new self-organizing organoid system that can easily be generated from the muscle layers of the chick fetal hindgut to study the emergence of spontaneous smooth muscle cell contractility. A key strength of this approach is that the organoids seem to contain few cell types (though more validation is needed), namely smooth muscle cells (SMCs) and interstitial cells of Cajal (ICCs). These organoids are amenable to live imaging of calcium dynamics as well as pharmacological perturbations for functional assays, and since they are derived from developing tissues, the emergence of the interactions between cell types can be functionally studied. Thus, the gut contractile organoids represent a reductionist system to study the interactions between SMCs and ICCs in comparison to the more complex in vivo environment, which has made studying these interactions challenging.

Thank you very much for the constructive comments.

Weaknesses:The study falls short in the sense that it does not provide a rigorous amount of evidence to validate that the gut organoids are made of bona fide smooth muscle cells and ICCs. For example, only two "marker" proteins are used to support the claims of cell identity of SMCs and ICCs. At the same time, certain aspects of the data are not quantified sufficiently to appreciate the variance of organoid rhythmic contractility. For example, most contractility plots show the trace for a single organoid. This leads to a concern for how reproducible certain aspects of the organoid system (e.g. wavelength between contractions/rhythm) might be, or how these evolve uniquely over time in culture. Furthermore, while this study might be able to capture the emergence of ICC-SMC interactions as they related to muscle contraction and pacemaking, it is unclear how these interactions relate to adult gastrointestinal physiology given that the organoids are derived from fetal cells that might not be fully differentiated or might have distinct functions from the adult. Finally, despite the strength of this system, discoveries made in it will need to be validated in vivo.Thank you very much for the comments, which are helpful to improve our MS. In the revised version, we have additionally used antibody against desmin, known to be a maker for mature SMCs (new Fig 3B). The signal is seen only in the peripheral cells overlapping with the αSMA staining (line 169-170).

Concerning the reproducibility, while contractility changes were shown for a representative organoid in the original version, experiments had been carried out multiple times, and consistent data were reproduced as already mentioned in the text of the first version of MS. However, we agree with this reviewer that it must be more convincing if we assess quantitatively. We have therefore conducted quantitative assessments of organoidal contractions and Ca^2+^ transients (new Fig. 2B, new Fig. 4D, new Fig 5D, E, new Fig. 6B, new Fig. 7B, new Fig. 8C, new Fig. S2, S3). Details such as repeats of experiments and size of specimens are carefully described in the revised version (Figure legends)

In particular, in place of contraction numbers/time, we have plotted “contraction intervals” between two successive peaks (Fig. 2B and others). Actually, with your suggestion, we have tried to perform a periodicity analysis of organoid contractions. Unfortunately, no clear value has been obtained, probably because the contractions/Ca^2+^ transitions are not as “regularly periodical” as seen in conventional physics. This led us to perform the peak-interval analysis. Methods to quantify the contraction intervals are carefully explained in the revised version.

As already mentioned in the “Our provisional responses” following the receipt of Reviewers’ comments, we agree that our organoids derived from embryonic hind gut (E15) might not necessarily recapitulate the full function of cells in adult. However, it has well been accepted in the field of developmental biology that studies with embryonic tissue/cells make a huge contribution to unveil complicated physiological cell functions. Nevertheless, we have carefully considered in the revised version so that the MS would not send misleading messages. We agree that in vivo validation of our gut contractile organoid must be wonderful, and this is a next step to go.

**Reviewer #3 (Public Review):**
Summary:The paper presents a novel contractile gut organoid system that allows for in vitro studying of rudimentary peristaltic motions in embryonic tissues by facilitating GCaMPlive imaging of Ca^2+^ dynamics, while highlighting the importance and sufficiency of ICC and SMC interactions in generating consistent contractions reminiscent of peristalsis. It also argues that ENS at later embryonic stages might not be necessary for coordination of peristalsis.Strengths:The manuscript by Yagasaki, Takahashi, and colleagues represents an exciting new addition to the toolkit available for studying fundamental questions in the development and physiology of the hindgut. The authors carefully lay out the protocol for generating contractile gut organoids from chick embryonic hindgut, and perform a series of experiments that illustrate the broader utility of these organoids for studying the gut. This reviewer is highly supportive of the manuscript, with only minor requests to improve confidence in the findings and broader impact of the work. These are detailed below.

Thank you very much for the constructive comments.

Weaknesses:(1) Given that the literature is conflicting on the role GAP junctions in potentiating communication between intestinal cells of Cajal (ICCs) and smooth muscle cells (SMCs), the experiments involving CBX and 18Beta-GA are well-justified. However, because neither treatment altered contractile frequency or synchronization of Ca++ transients, it would be important to demonstrate that the treatments did indeed inhibit GAP junction function as administered. This would strengthen the conclusion that GAP junctions are not required, and eliminate the alternative explanation that the treatments themselves failed to block GAP junction activity.

Thank you for these comments, and we agree. In the revised version, we have verified the drugs, CBX and 18b-GA, using dissociated embryonic heart cells in culture, a well-established model for the gap junction study (new Fig. S3D, line 237-239). Expectedly, both inhibitors abrogate the rhythmic beats of heart cells, and importantly, cells’ beats resume after wash-out of the drug.

(2) Given that 5uM blebbistatin increases the frequency of contractions but 10uM completely abolishes contractions, confirming that cell viability is not compromised at the higher concentration would build confidence that the phenotype results from inhibition of myosin activity. One could either assay for cell death, or perform washout experiments to test for recovery of cyclic contractions upon removal of blebbistatin. The latter may provide access to other interesting questions as well. For example, do organoids retain memory of their prior setpoint or arrive at a new firing frequency after washout?

We greatly appreciate these suggestions and also interesting ideas to explore! In the revised version, we have newly conducted washout experiments (new Fig. 6B) (10 µM drug is washed-out from culture medium), and found that contractions resume, showing that cell viability is not compromised at 10 µM concentration (line 257-259). Intriguingly, the resumed rhythm appears more regular than that before drug administration. Thus, the contraction rhythm of the organoid might be determined by cellcell interactions at any given time rather than by memory of their prior setpoint. This is an interesting issue we would like to further explore in the future. These issues, although potentially interesting, are not mentioned in the text of the revised version, since it is too early to interpret there observations.

(3) Regulation of contractile activity was attributed to ICCs, with authors reasoning that Tuj1+ enteric neurons were only present in organoids in very small numbers (~1%).However, neuronal function is not strictly dependent on abundance, and some experimental support for the relative importance of ICCs over Tuj1+ cells would strengthen a central assumption of the work that ICCs the predominant cell type regulating organoid contraction. For example, one could envision forming organoids from embryos in which neural crest cells have been ablated via microdissection or targeted electroporation. Another approach would be ablation of Tuj1+ cells from the formed organoids via tetrodotoxin treatment. The ability of organoids to maintain rhythmic contractile activity in the total absence of Tuj1+ cells would add confidence that the ICCs are indeed the driver of contractility in these organoids.

We agree. In the revised version, we have conducted TTX administration (new Fig. S2C). Changes in contractility by this treatment is not detected, supporting the argument that neural cells/activities are not essential for rhythmic contractions of the organoid (line 178-181).

(4) Given the implications of a time lag between Ca++ peaks in ICCs and SMCs, it would be important to quantify this, including standard deviations, rather than showing representative plots from a single sample.

In the revised version, we have elaborated a series of quantitative assessments as mentioned above (please see our responses to the “eLife assessments” at the beginning of these correspondences). The latency between Ca^2+^ peaks in ICCs and SMCs is shown in new Fig. 4D, in which measured value is 700 msec-terraced since the time-lapse imaging was performed with 700 msec intervals (as already described in the first version).

117 peaks for 14 organoids have been assessed (line 218).

(5) To validate the organoid as a faithful recreation of in vivo conditions, it would be helpful for authors to test some of the more exciting findings on explanted hindgut tissue. One could explant hindguts and test whether blebbistatin treatment silences peristaltic contractions as it does in organoids, or following RCAS-GCAMP infection at earlier stages, one could test the effects of GAP junction inhibitors on Ca++ transients in explanted hindguts. These would potentially serve as useful validation for the gut contractile organoid, and further emphasize the utility of studying these simplified systems for understanding more complex phenomena in vivo.

Thank you very much for insightful comments. We would love to explore these issues in near future. Just a note is that it was previously reported that Nifedipine silences peristaltic contractions in ex-vivo cultured gut (Chevalier et al., 2024; Der et al., 2000; Der-Silaphet et al., 1998).

(6) Organoid fusion experiments are very interesting. It appears that immediately after fusion, the contraction frequency is markedly reduced. Authors should comment on this, and how it changes over time following fusion. Further, is there a relationship between aggregate size and contractile frequency? There are many interesting points that could be discussed here, even if experimental investigation of these points is left to future work.

It would indeed be interesting to explore how cell communications affect/determine the contraction rhythm, and our novel organoids must serve as an excellent model to address these fundamental questions. We have observed multiple times that when two organoids fuse, they undergo “pause”, and resume coordinated contractions as a whole, and we have mentioned such notice briefly in the revised version (line 282). To know what is going on during this pause time should be tempting. In addition, we have an impression that the larger in size organoids grow, the slower rhythm they count. We would love to explore this in near future.

(7) Minor: As seen in Movie 6 and Figure 6A, 5uM blebbistatin causes a remarkable increase in the frequency of contractions. Given the regular periodicity of these contractions, it is a surprising and potentially interesting finding, but authors do not comment on it. It would be helpful to note this disparity between 5 and 10 uM treatments, if not to speculate on what it means, even if it is beyond the scope of the present study to understand this further.

We assume that the increase in the frequency of contractions at 5 µM might be due to a shorter refractory period caused by a decreasing magnitude (amplitude) of contraction. We have made a short description in the revised text (line 256-257).

(8) Minor: While ENS cells are limited in the organoid, it would be helpful to quantify the number of SMCs for comparison in Supplemental Figure S2. In several images, the number of SMCs appears quite limited as well, and the comparison would lend context and a point of reference for the data presented in Figure S2B.

In the revised version, the number of SMCs has been counted and added in Fig. S2B. Contrary to that SMCs are more abundant than ICCs in an intact gut, the proportion is reversed in our organoid (line 181-183). It might due to treatments during cell dissociation/plating.

(9) Minor: additional details in the Figure 8 legend would improve interpretation of these results. For example, what is indicated in orange signal present in panels C, G and H? Is this GCAMP?

We apologize for this confusion. In the revised version, we have added labeling directly in the photos of new Fig. 9 (old Fig. 8). For C, G and H, the left photo is mRuby3+GCaMP6s, and the right one is GCaMP6s only.

**Recommendations for the authors:**

**Reviewer #1 (Recommendations For The Authors):**
I have a few comments for the authors to consider:(1) Figure 4C: The authors propose that calcium signals propagate from ICC to SMC based on the results presented in this figure. While it is observed that the peak of the calcium signal in ICC precedes that in SMC, it's worth noting that the onset of the rise in calcium signals occurs simultaneously in ICC and SMC. Doesn't this suggest that they are activated simultaneously? The latency observed for the peaks of calcium signals could reflect different kinetics of the rise in calcium concentration in the two types of cells rather than the order of calcium signal propagation.

We greatly appreciate these comments. We have re-examined kinetics of GCaMP signals in ICC and SMC, but we did not succeed in validating rise points precisely. We agree that the possibility that the rise in calcium signals could be occurring simultaneously. To clarify these issues, analyses with higher resolution is required, such as using GCaMP6f or GCaMP7/8. Nevertheless, the disappearance of the latency of Ca^2+^ peak by CBX implies a role of gap junction in ICC to SMC signaling. In the revised version, we replaced the wording “rise” by “peak” when the latency is discussed.

(2) Figure 5C: The specific elimination of the latency in the calcium signal peaks between ICC and SMC is interesting. However, I am curious about how gap junction inhibitors specifically eliminate the latency between ICC and SMC without affecting other aspects of calcium transients in these cells, such as amplitude and synchronization among ICCs and/or SMCs. Readers of the manuscript would expect some discussion on possible mechanisms underlying this specificity. Additionally, I wonder if the elimination of the latency was observed consistently across all samples examined. The authors should provide information on the frequency and number of samples examined, and whether the elimination occurs when 18-beta-GA is used.

In the revised version, we have elaborated quantitative demonstration. For the effects by CBX on latency or Ca^2+^ peaks, a new graph has been added to new Fig 5, in which 100 µM eliminated the latency. Intriguingly, the latency appears to be attributed to a gap junction that is not inhibited by18-beta-GA (please see new Fig. S3E). As already mentioned above, inhibiting activity of both CBX and 18-beta-GA has been verified using dissociated cells of embryonic heart, a popular model for gap junction studies.

At present, we do not know how gap junction(s) contribute to the latency of Ca^2+^ peaks without affecting synchronization among ICCs and/or SMCs (we have not addressed amplitude of the oscillation in this study). Actually, it was surprising to us to find that GJ’s contribution is very limited. We do not exclude the importance of GJs, and currently speculate that GJs might be important for the initiation of contraction/oscillation signals, whereas the requirement of GJs diminishes once the ICC-SMC interacting rhythm is established. What we observed in this study might be the synchronization signals AFTER these interactions are established (Day 7 of organoidal culture). Upon the establishment, it is possible that mechanical signaling elicited by smooth muscles’ contraction might become prominent as a mediator for the (stable) synchronization, as implicated by experiments with blebbistatin and Nifedipin, the latter being newly added to the revised version (new Fig. 7). We have added such speculation, although briefly in Discussion (line 374-377)

(3) Figure 6: The significant effects of blebbistatin on calcium dynamics in both ICC and SMC are intriguing. However, since only one blocker is utilized, the specificity of the effects is unclear. If other blockers for muscle contraction are available, they should be employed. Considering that a rise in calcium concentration precedes contraction, calcium transients should persist even if muscle contraction is inhibited. One concern is whether blebbistatin inadvertently rendered the cells unhealthy. The authors should demonstrate at least that contraction and calcium transients recover after removal of the drug. The frequency and number of samples examined should be shown, as requested for Figure 5C above.

Thank you for these critical comments. A possible harmfulness of the drugs was also raised by other reviewers, and we have therefore conducted wash-out experiments in the revised version (new Fig. 6B). Contractions resume after wash-out showing that cell viability is not compromised at 10 µM concentration. The number of samples examined has been described more explicitly in the revised version. Regarding the blocker of SMC, we have newly carried out pharmacological assays using nifedipine, a blocker of a L-type Ca^2+^ channel known to operate in smooth muscle cells (new Fig 7) (Chevalier et al., 2024; Der et al., 2000; Der-Silaphet et al., 1998). As already explained in the “Responses to eLife assessment”, the treatment abrogated ICCs’ rhythm and synchronous Ca^2+^ transients between ICCs and SMCs, further corroborating our model that not only ICC-to-SMC interactions but also SMC-to-ICC feedback signals are operating to achieve coordinated/stable rhythm of gut contractile organoids of Day 7 culture (please also see our responses shown above for Comment (2)).

**Reviewer #2 (Recommendations For The Authors):**
Major:(1) The claim that organoids contain functional SMCs and ICCs is insufficient as it currently relies on only c-Kit and aSMA antibodies. This conclusion could be additionally supported by staining with other markers of contractile smooth muscle (e.g. TAGLN and MYH14) and an additional accepted marker of ICCs (e.g. ANO1/TMEM16). Moreover, it should be demonstrated whether these cells are PDGFRA+, as PDGFRA is a known marker of other mesenchymal fibroblast cell types. These experiments would additionally rule out whether these cells were simply less differentiated myofibroblasts. Given that there might not be available antibodies that react with chicken protein versions, the authors could support their conclusions using alternative approaches, such as fluorescent in situ hybridization. A more thorough approach, such as single-cell RNA sequencing to compare the cell composition of the in vitro organoids to the in vivo colon, would fully justify the use of these organoids as a system for studying in vivo cell physiology.

With these suggestions provided, we have newly stained contractile organoids with anti-desmin antibody, known to be a marker for differentiated SMCs. As shown in new Fig. 3B, desmin-positive cells perfectly overlapped with aSMA-staining, indicating that the peripherally enclosing cells are SMCs. Regarding the interior cells, as this Reviewer concerned, there are no antibodies against ANO1/TMEM16 which are available for avian specimens. The anti- c-Kit antibody used in this study is what we raised in our hands by spending years (Yagasaki et al., 2021), in which the antibody was carefully validated in intact guts of chicken embryos by multiple methods including Western Blot analyses, immunostaining, and in situ hybridization. We have attempted several times to perform organoidal whole-mount in situ hybridization for expression of PDGFRα, but we have not succeeded so far. In addition, as explained to the Editor, the very unhealthy condition of purchased eggs these past 7 months did not allow us to continue any further. We are planning to interrogate cell types residing in the central area of the organoid, results of which will be reported in a separate paper in near future.

(2) The key ICC-SMC relationship and physiological interaction seems to arise developmentally, but the mechanisms of this transition are not well defined (Chevalier 2020). To further support the claim that ICC-SMC interactions can be interrogated in this system, this study would benefit from establishing organoids at distinct developmental stages to (a) show that they have unique contractile profiles, and (b) demonstrate that they evolve over time in vitro toward an ICC-driven mechanism.

We agree with these comments. We tried to prepare gut contractile organoids derived from different stages of development, and we had an impression that slightly younger hindguts are available for the organoid preparations. In addition, not only the hindgut, but also midgut and caecum also yield organoids. However, since formed organoids derived from these “non-E15 hindgut” vary substantially in shapes, contraction frequencies/amplitudes etc., we are currently not ready to report these preliminary observations. Instead, we decided to optimize and elaborate in vitro culture conditions by focusing on the E15 hindgut, which turned out to be most stable in our hands. Nevertheless, it is tempting to see how organoid evolves over time during gut development.

(3) This manuscript would be greatly enhanced by a functional examination of the prospective organoid ICCs. For example, the authors could test whether the c-Kit inhibitor Imatinib, which has previously been used to impair ICC differentiation and function in the developing chick gut (Chevalier 2020), has an effect on contractility at different stages.

Following the paper of (Chevalier 2020), we had already conducted similar experiments with Imatinib in the culture with our organoids, but we did not see detectable effects. In that paper, the midgut of younger embryos was used, whereas we used E15 hindgut to prepare organoids. It would be interesting to see if we add Imanitib earlier during organoidal formation, and this is a next step to go.

(4) It is claimed that there is a 690s msec delay in SMC spike relative to ICC spike, however, it is unclear where this average is derived from and whether the organoid calcium trace shown in Figure 4C is representative of the data. The latency quantification should be shown across multiple organoids, and again in the case of carbenoxolone treatment, to better understand the variations in treatment.

We apologize that the first version failed to clearly demonstrate quantitative assessments. In the revised version, we have elaborated quantitative assessments (117 peaks for 14 organoids) (line 216-218). In new Fig. 4D, measured value is 700 msecterraced since as already mentioned in the first version, the time-lapse imaging was performed with 700 msec intervals.

(5) As above, a larger issue is that only single traces are shown for each organoid. This makes it challenging to understand the variance in contractile properties across multiple organoids. While contraction frequencies are shown several times, the manuscript would benefit from additional quantifications, such as rhythm (average wavelength between events) in control and perturbed conditions.

We have substantially elaborated quantitative assessments (please also see our responses to the “Public Review”). In particular, in place of contraction numbers/time, we have plotted “contraction intervals” between two successive peaks (Fig. 2B and others). Actually, we have tried to perform a periodicity analysis of organoid contractions. Unfortunately, no clear value has been obtained, probably because the contractions/Ca^2+^ transitions are not as “regularly periodical” as seen in conventional physics. This led us to perform the peak-interval analysis. Methods to quantify the contraction intervals are carefully explained in the revised version.

(6) The synchronicity observed between ICCs and SMCs within the organoid is interesting, and should be emphasized by making analyses more quantitative so as to understand how consistent and reproducible this phenomenon is across organoids. Moreover, one of the most exciting parts of the study is the synchronicity established between organoids in the hydrogel system, but it is insufficiently quantified. For example, how rapidly is pacemaking synchronization achieved?

As we replied above to (5), and described in the responses to the “Public Review”, we have substantially elaborated quantitative assessments in the revised version. Concerning the synchronicity between ICCs and SMCs, our data explicitly show that as long as the organoid undergoes healthy contraction, they perfectly match their rhythm (Fig. 4) making it difficult to display quantitatively. Instead, to demonstrate such synchronicity more convincingly, we have carefully described the number of peaks and the number of independent organoids we analyzed in each of Figure legends. In the experiments with hydrogels, the time required for two organoids to start/resume synchronous contraction varies greatly. For example, for the experiment shown in new Fig 9F, it takes 1 day to 2 days for cells crawling out of organoids and cover the surface of the hydrogel. In the experiments shown in new Fig. 8, two organoids undergo “pause” before resuming contractions. In the revised version, we have briefly mentioned our notice and speculation that active cell communications take place during this pausing time, (line 282-283 in Result and line 437-439 in Discussion). We agree with this reviewer saying that the pausing time is potentially very interesting. However, it is currently difficult to quantify these phenomena. More elaborate experimental design might be needed.

(7) Smooth muscle layers in vivo are well organized into circular and longitudinal layers. To establish physiological relevance, the authors should demonstrate if these organoids have multiple layers (though it looks like just a single outer layer) and if they show supracellular organization across the organoid.

The immunostaining data suggest that peripherally lining cells are of a single layer, and we assume that they might be aligned in register with contracting direction. However, to clarify these issues, observation with higher resolution would be required.

(8) To further examine whether the organoids contain true functional ICCs, the authors should test whether their calcium transients are impacted by inhibitors of L-type calcium channels, such as nifedipine and nicardipine. These channels have been demonstrated to be important for SMCs but not ICCs, so one might expect to see continued transients in the core ICCs but a loss of them in SMCs (Lee et al., 1999; PMID: 10444456)

We appreciate these comments. We have accordingly conducted new experiments with Nifedipine. Contrary to the expectation, Nifedipine ceases not only organoidal contractions, but also ICC activities (and its resulting synchronization) (new Fig. 7). These findings actually corroborate our model already mentioned in the first version that ICCs receive mechanical feedback from SMC’s contraction to stably maintain their oscillatory rhythm. We believe that the additional findings with Nifedipine have improved the quality of our paper. Concerning the central cells in the organoid, we have additionally used anti-desmin antibody known to mark differentiated SMCs. Desmin signals perfectly overlap with those of aSMA in the peripheral single layer, supporting that the peripheral cells are SMCs and central cells are ICCs. The anti c-Kit antibody used in this study is what we raised in our hands by spending years (Yagasaki et al., 2021), in which the antibody was carefully validated in intact guts of chicken embryos by multiple methods including Western Blot analyses, immunostaining, and in situ hybridization.

ANO1/TMEM16 are known to stain ICCs in mice. Antibodies against ANO1/TMEM16 available for avian specimens are awaited.

(9) Despite Tuj1+ enteric neurons only making up a small fraction of the organoids, the authors should still functionally test whether they regulate any aspect of contractility by treating organoids with an inhibitor such as tetrodotoxin to rule out a role for them.

Thank you for these advices, which are also raised by other reviewers. We have conducted TTX administration (new Fig. S2C). Changes in contractility by this treatment is not detected, supporting the argument that neural cells/activities are not essential for rhythmic contractions of the organoid (line 178-181).

(10) Finally, the manuscript is written to suggest that the focus of the study is to establish a system to interrogate ICC-SMC interactions in gut physiology and peristalsis. However, the organoids designed in this study are derived from the fetal precursors to the adult cell types. Thus, they might not accurately portray the adult cell physiology. I don't believe that this is a downfall, but rather a strength of the study that should be emphasized. That is, the focus could be shifted toward stressing the power of this new system as a reductionist, self-organizing model to examine the developmental emergence of contractile synchronization in the intestine - in particular that arising through ICC-SMC interactions.

We appreciate these advices. In the revised MS, we are careful so that our findings do not necessarily portray the physiological functions in adult gut.

Minor:More technical information could be used in the methods:(1) What concentration of Matrigel is used for coating, and what size were the wells that cells were deposited into?

We have added, “14-mm diameter glass-bottom dishes (Matsunami, D11130H)” and “undiluted Matrigel (Corning, 354248) at 38.5°C for 20 min” (line 471473).

(2) How were organoids transferred to the hydrogels? And were the hydrogels coated?

We have added “Organoids were transferred to the hydrogel using a glass capillary” (line 560-561).

(3) Tests for significance and p values should be added where appropriate (e.g. Figure S3B).

We have added these in Figure legend of new Fig. S3.

**Reviewer #3 (Recommendations For The Authors):**
This is an exciting study, and while the majority of our comments are minor suggestions to improve the clarity and impact of findings, it would be important to verify the effective disruption of GAP junction function with CBX or 18Beta-GA treatments before concluding they are not required for coordination of contractility and initiation by ICCs. It is possible that sufficient contextual support exists in the literature for the nature of treatments used, but this may need to be conveyed within the manuscript to allay concerns that the results could be explained by ineffective inhibition of GAP junctions.

Thank you very much for these advices. In the revised version, we have newly carried out experiments with dissociated embryonic heart cells cultured in vitro, a model widely used for gap junction studies (Fig. S3D). Both CBX or 18b-GA exert efficient inhibiting activity on contractions of heart cells. We have added the following sentence, “The inhibiting activity of the drugs used here was verified using embryonic heart culture (line 237-239)”.